# LONG-TERM ACTION ANTICIPATION VIA TRANSCRIPT-BASED SUPERVISION

## ABSTRACT

Long-Term Anticipation (LTA) from video is a crucial task in computer vision, with significant implications for human-machine interaction, robotics, and beyond. However, to date, it has been tackled exclusively in a fully supervised manner, by relying on dense frame-level annotations that hinder scalability and limit real-world applicability. To address this limitation, we introduce TbLTA (Transcript-based LTA), the first weakly-supervised approach for LTA, which relies solely on video transcripts during training. This high-level semantic supervision provides the narrative temporal structure that can guide the model toward understanding the relationships between events over time. Our model is built on an encoder-decoder architecture, which is trained using dense pseudo-labels generated by a temporal alignment module to supervise the predictions of both the segmentation head and the anticipation decoder. In addition, the video transcript itself is also used for 1) enhancing video features by contextually grounding them through cross-modal attention, 2) supplying a more global supervision to the model action segmentation predictions over the full video, which in turn helps to provide a better contextualized representation to the anticipation decoder. Through experiments on the Breakfast, 50Salads, and EGTEA benchmarks, we demonstrate that transcript-based supervision offers a very robust and less costly alternative to its fully supervised counterpart for the LTA task [1].

## 1 INTRODUCTION

Understanding and anticipating human actions in videos is a fundamental capability for intelligent systems operating in dynamic environments (He et al., 2024; Dalal et al., 2025). In particular, the task of Long-Term Action Anticipation (LTA) aims to predict future actions several minutes ahead based on partial observations. Extracting meaningful information from such observations typically requires segmenting them into temporally aligned action labels, a task known as Temporal Action Segmentation (TAS).

Recent approaches for both TAS and LTA have achieved substantial progress by leveraging dense annotations (Gong et al., 2022b; Abu Farha et al., 2018; Gong et al., 2024; Nawhal et al., 2022b; Zhong et al., 2023a; Huang et al., 2025; Lu & Elhamifar, 2024; Bahrami et al., 2023). However, highly granular labeling is costly and difficult to scale, especially for long and fine-grained activity sequences. While recent efforts in TAS have increasingly embraced weakly-/unsupervised settings (Xu & Zheng, 2024; Zhang et al., 2023; Bueno-Benito & Dimiccoli, 2025; Xu & Gould, 2024; Spurio et al., 2024), LTA remains largely unexplored under weak / no supervision. The only attempt to address the annotation burden for LTA was proposed in Zhang et al. (2021). It combines a small set of fully labeled sequences with weak labels for the next action, using pseudo-label refinement to approximate future boundaries. Yet this approach still relies on temporally localized human annotations, which have a narrow focus on the present and lack a high-level temporal understanding.

In this work, we propose **TbLTA**, the first weakly-supervised LTA model trained exclusively with *video transcripts*—an ordered action list, without timing or duration information—which are significantly cheaper to obtain with respect to dense annotations. Since LTA is about understanding the logical progression of steps within a larger activity being performed, transcripts, with the power of semantic abstraction, are specially suited to this task. In addition to the supervision provided by the transcripts themselves, we explicitly temporally align action labels with the video sequences

---

[1]Code will be released in a *GitHub* repository upon acceptance.

Figure 1: Given video features and transcripts, TbLTA aligns the transcript to the video through a temporal alignment module, producing frame-level pseudo-labels for supervision. During training, the transcript further provides global guidance via a dedicated loss and enriches video features through cross-modal attention, enabling dense anticipation without frame-level annotations.

through a dedicated temporal alignment module, and we use the generated pseudolabels for frame-level supervision (see Figure 1). Furthermore, we leverage the transcripts to enrich video features by contextually grounding them with verbs and objects appearing on it through a cross-modal attention layer. Finally, following previous work Gong et al. (2024), we segment the full video during training instead of just the observation interval, to ensure that the decoder can learn long-range temporal dependencies occurring after the observation ends. Our main contributions are:

- We propose for the first time to train a model for LTA by using only video transcripts without boundary annotations as supervision.

- We introduce **TbLTA**, a novel encoder-decoder architecture for LTA transcript-based supervision, where the encoder learns to capture fine-grained long-range temporal relations between all frames of the video, and the decoder learns to capture global relations occurring after the observation ends, along with the observed features from the encoder.

- We propose to temporally align video transcripts to frame-level features and leverage the estimated pseudo-labels for supervising both segmentation and anticipation.

- We leverage transcripts not only as weak supervision, but also as semantic context to enrich video features through a dedicated cross-modal attention.

- We establish the first transcript-only supervision baseline for LTA on Breakfast (Kuehne et al., 2014), 50Salads (Stein & McKenna, 2013), and EGTEA (Li et al., 2018), showing that weak supervision can yield competitive long-horizon anticipation.

## 2 RELATED WORK

**Temporal Action Segmentation (TAS)** aims to assign an action label to every frame of long, untrimmed videos, producing coherent segments with accurate boundaries. Approaches are typically grouped by supervision level. Fully supervised methods achieve the most reliable performance but require dense frame-level annotations (Liu et al., 2023; Huang et al., 2025; Bahrami et al., 2023; Behrmann et al., 2022; Aziere et al., 2025). To improve generalization and scalability, recent research has shifted toward semi/weakly-supervised (Xu & Zheng, 2024; Zhang et al., 2023) and unsupervised paradigms (Li et al., 2024; Xu & Gould, 2024; Spurio et al., 2024), which reduce reliance on exhaustive annotations while maintaining competitive accuracy. Advances include weakly-supervised methods that mitigate noisy boundaries using transcript-level supervision and video-level regularization (Xu & Zheng, 2024), and unsupervised approaches such as CLOT (Bueno-Benito & Dimiccoli, 2025), which introduces an OT-based framework with multi-level cyclic feature learning to enforce segment-level consistency and improve generalization.

**Action Anticipation** has been widely studied under different conditions, varying in observable inputs, temporal horizons, and action granularity. The goal is to infer future actions from observed video data, with existing works addressing this through diverse formulations such as predicting the next action and its start time (Zhong et al., 2023a; Thakur et al., 2024; Zhang et al., 2024a), inferring the final goal (Wang et al., 2023), or planning procedural steps (Surís et al., 2021). Based on the prediction horizon (Zhong et al., 2023b), methods are broadly divided into short-term and long-term anticipation. Short-term approaches focus on predicting actions a few seconds ahead using low-level cues (Guo et al., 2024; Diko et al., 2024), whereas *long-term anticipation* (LTA) forecasts sequences of actions over extended horizons, facing challenges such as long-range dependency modeling, autoregressive error accumulation, and the uncertainty of plausible futures (Lai et al., 2024).

**Long-term Action Anticipation (LTA)** focuses on forecasting sequences of future actions over extended temporal horizons, has seen rapid progress across a variety of modeling paradigms. Early works framed LTA as a duration-agnostic transcript prediction problem, often adopting transformer-based architectures (Nawhal et al., 2022b). More recent approaches have incorporated object-centric representations (Zhang et al., 2024b), integrated large language and video–language models (Zhao et al., 2024; Mittal et al., 2024). In particular, Kim et al. (2024) explored language-based anticipation without explicit time annotations, using a vision–language model with in-context learning and MMR to predict symbolic sequences of future actions. Within this landscape, we focus on the task of *dense long-term action anticipation*, where the aim is to generate frame-level forecasts of future actions for a predefined number of upcoming frames. The task of dense anticipation was first introduced by Abu Farha et al. (2018) and propose two models (RNN and CNN), Abu Farha & Gall (2019) introduces a GRU network to model the uncertainty of future activities in an autoregressive way, and Sener et al. (2020) proposes TempAgg, an end-to-end model, employing the action segmentation model for visual features in training with cycle consistency between past and future actions. Abu Farha et al. (2020a) suggests a multi-scale temporal aggregation model that pools past visual features in condensed vectors and then iteratively predicts future actions using the LSTM network. More recent contributions can be broadly divided into *deterministic* approaches, which output a single most likely future, and *stochastic* approaches, which explicitly model uncertainty by generating multiple plausible futures. Deterministic models include FUTR (Gong et al., 2022b), which anticipates all future actions in parallel from fine-grained past features, and ANTICIPATR (Nawhal et al., 2022b) which uses a two-stage training pipeline. On the other hand, stochastic methods have leveraged diffusion-based generative modeling (Zatsarynna et al., 2024; 2025), producing diverse yet consistent future sequences. A notable extension is Actfusion (Gong et al., 2024), which unifies TAS and LTA into a diffusion-based framework.

Despite these advances, most dense anticipation methods still depend on costly frame-level annotations. Zhang et al. (2021) took a step forward by exploring a method both semi- and weakly-supervised for dense LTA, where a small set of fully-labelled data together with weak labels is used for supervision. In the weakly annotated part of the data, the video segment is annotated only with the first action class of the anticipated sequence, instead of all frames in the sequence. In contrast, we completely eliminate dense annotations and propose **TbLTA**, the first fully weakly-supervised framework for dense LTA, trained exclusively from transcripts (ordered action lists without timing or duration), thereby avoiding expensive boundary labels.

**Sequence-to-sequence modeling in video understanding.** A substantial body of prior work addresses sequence-to-sequence alignment between video frames and action transcripts through the use of structured objectives. Classical approaches include Hidden Markov Models (HMMs) with Viterbi decoding, originally inspired by speech recognition, to capture action–frame transitions under weak supervision (Kuehne et al., 2016). Similarly, Dynamic Time Warping (DTW) has long been applied for temporal alignment and was recently revisited in a differentiable form to enable end-to-end optimization (Chang et al., 2021). The Connectionist Temporal Classification (CTC) loss (Graves, 2012) has been extensively adopted in sequence-to-sequence learning, particularly when frame-level annotations are unavailable. Its application to weakly-supervised action segmentation was pioneered by Huang et al. (2016), who proposed ECTC to enforce alignments consistent with visual similarities. Building on this, Ng & Fernando (2021) combined CTC with attention to better exploit transcript-level supervision. While these works primarily target segmentation, we extend the use of CTC-style objectives to the task of *dense long-term anticipation*, demonstrating that transcript-only supervision can drive frame-level forecasting without costly boundary annotations. In parallel, Conditional Random Fields (CRFs) further extended these ideas by modeling richer temporal dependencies in sequence prediction (Huang et al., 2015; Mavroudi et al., 2018). More recently, Maté & Dimiccoli (2024) introduced a CRF formulation specifically for long-term anticipation (LTA). While their approach is deterministic, we propose a stochastic variant that explicitly captures the uncertainty inherent in LTA predictions.

## 3 METHODOLOGY

***Problem Definition.***

We address the task of dense long-term action anticipation under weak supervision, where training relies solely on transcripts that always refer to an action-sequence transcript, i.e., an ordered list of action labels, without providing frame-level temporal annotations, boundaries, or durations.

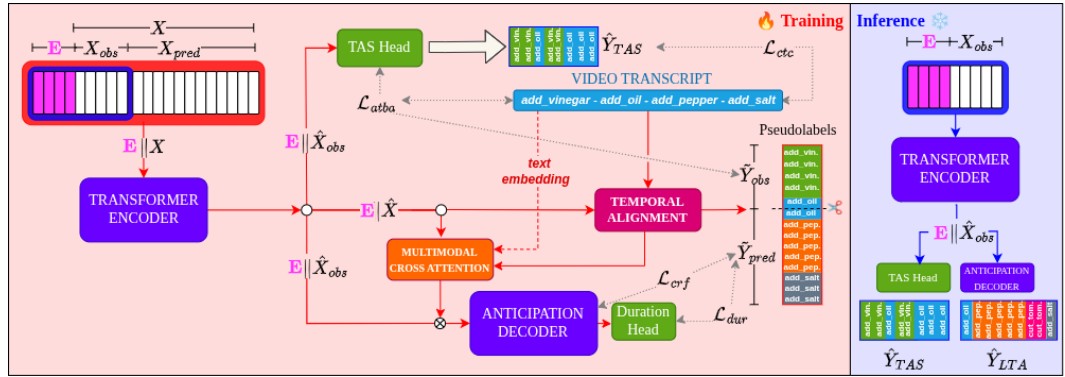

Figure 2: **Overview of the proposed TbLTA framework**. During training, the model takes as input video features and the corresponding video transcript ($X = [X_{obs}, X_{pred}], \mathcal{Y}$), and generates dense pseudo-labels for the full video $\hat{Y} = [\hat{Y}_{obs}, \hat{Y}_{pred}]$. These pseudolabels are used: 1) to supervise the prediction of action segmentation labels $\hat{Y}_{TAS}$ on the full video and of action anticipation labels $\hat{Y}_{LTA}$ in the anticipation interval through multiple cross-entropy losses; 2) to build an attention mask for cross-modal attention, ensuring that text embeddings attends only to the most aligned video segments rather than the entire sequence, with the goal of contextually grounding video features. The video transcript is also used to supervise globally the TAS predictions through a CTC loss $\mathcal{L}_{ctc}$.

Formally, a video is represented as a sequence of $T$ frames with associated feature vectors $X = \{x_1, x_2, \ldots, x_T\} \in \mathbb{R}^{T \times d}$, where $x_t \in \mathbb{R}^d$ denotes the feature vector extracted from $t$-th frame and $d$ dimension of embedding. Let $\alpha, \beta \in (0, 1)$ with $\alpha + \beta \leq 1$. The observed temporal features are $X_{obs} = \{x_1, \ldots, x_{\lfloor \alpha T \rfloor}\}$ and $X_{pred} = \{x_{\lfloor \alpha T \rfloor + 1}, \ldots, x_{\lfloor (\alpha + \beta) T \rfloor}\}$, with lengths $T_{obs} = \lfloor \alpha T \rfloor$ and $T_{pred} = \lfloor \beta T \rfloor$. Each video is annotated with a transcript $\mathcal{Y} = [y_1, \ldots, y_N]$, where $y_n \in \mathcal{C}$, $\mathcal{C}$ is the action vocabulary, and $N$ is the number of action segments in the video.

In a weakly-supervised setting, $\mathcal{Y}$ and $X$ are not temporally aligned. To address this, we introduce a set of learnable class tokens $E \in \mathbb{R}^{|\mathcal{C}| \times d}$, which serve as latent action prototypes. During training, the input to the model is the concatenation of video features and class tokens, i.e. $[E \parallel X]$, allowing the encoder to jointly reason over visual evidence and class-level priors. At inference, only $[E \parallel X_{obs}]$ is provided. The objective is twofold: (1) during training, by using $[E \parallel X]$ and the transcript $\mathcal{Y}$, the model must align the continuous feature sequence with the discrete ordered list of actions, and generate frame-level pseudo-labels $\hat{\mathcal{Y}} = [\hat{\mathcal{Y}}_{obs}, \hat{\mathcal{Y}}_{pred}]$ for the full video. These pseudo-labels supervise the TAS head and the LTA decoder on the future interval; (2) At inference, given only the observed features $X_{obs}$ and learned class token $E$, the model predicts the sequence of future actions $\hat{\mathcal{Y}}_{LTA} = [\hat{y}_{k+1}, \ldots, \hat{y}_N]$ and their durations $D = [d_{k+1}, \ldots, d_N] \in \mathbb{R}^{N-k^*}$, with $\sum_{j=k+1}^{N} d_j = 1$ following Abu Farha et al. (2018), where $k^*$ denotes the (unknown) boundary index between observed and future actions. Since $k^*$ is not observed, the model must implicitly estimate both the boundary and the corresponding observed pseudolabels $\hat{\mathcal{Y}}_{obs} = [\hat{y}_1, \ldots, \hat{y}_{k^*}]$ by temporal alignment. Thus, the task is to learn a parametric function $f_\theta : \mathbb{R}^{T_{obs} \times d} \to (\hat{\mathcal{Y}}_{LTA}, \hat{D})$ that, given observed features, anticipates future actions and their durations while jointly inferring actions and their boundaries under weak supervision.

### 3.1 MODEL ARCHITECTURE

We propose **TbLTA**, illustrated in Fig.2, a modular transformer-based architecture designed for the LTA task and trained exclusively via video transcripts. The architecture consists of a transformer encoder, a weakly-supervised temporal alignment module, a cross-attention layer between video and transcript, a segmentation head, and an anticipation decoder.

**Transformer encoder.** The input video features $X$ are first projected to the model dimension and concatenated with a set of learnable class tokens $E$, which act as latent action prototypes. The resulting sequence is processed by a temporal network. Following prior work, we adopt a transformer encoder with learnable positional embeddings and a pyramid hierarchical local attention mechanism (Vaswani et al., 2017). To enable the decoder to acquire a comprehensive representation of the future's temporal structure, the encoder is trained over the entire video sequence. This design explicitly links the encoder's outputs to future actions, thereby strengthening the connection between past observations and anticipated events.

**Weakly-supervised temporal alignment module.** In the absence of frame-level annotations, our framework introduces an intermediate weakly-supervised temporal alignment stage to bridge the gap between symbolic transcripts and frame-level features. In practice, we adopt the ATBA module proposed in (Xu & Zheng, 2024) to partition the full transcript $\mathcal{Y}$ into observed and future sub-transcripts, $\mathcal{Y}_{\text{obs}}$ and $\mathcal{Y}_{\text{future}}$, corresponding to the observed and anticipated portions of the video. The advantage of ATBA is that it generates soft per-frame pseudo-labels that preserve boundary uncertainty, crucial for long-horizon anticipation, where hard labels are often unreliable near transitions. Jointly with the generated pseudo-labels $\hat{\mathcal{Y}}$, the temporal alignment module also contributes to learn a new encoding for the initial features that are more suited for the task of action anticipation.

**Segmentation head.** For the full video features $X$, a linear classifier predicts frame-level logits $\mathcal{Y}_{\text{TAS}}$. This module also stabilizes encoder representations for downstream anticipation.

**Cross-attention layer between modalities.** Transcripts are typically exploited only as sequence-level ordering constraints. In contrast, we explicitly couple them with video features through a *local cross-modal mechanism*. Let $A = [a_1, \ldots, a_N] \in \mathbb{R}^{N \times d}$ denote transcript embeddings, where each $a_i$ is obtained from a pre-trained language model applied to the natural-language action label (Sanh et al., 2019). Given encoder features $\hat{X} \in \mathbb{R}^{T \times d}$ and pseudo-labels $\hat{\mathcal{Y}}$, we construct a binary local mask $M \in \{0,1\}^{N \times T}$ that restricts each action $a_i$ to a temporal neighborhood around its predicted occurrence. Cross-attention is then defined as

$$A \leftarrow \text{softmax}\left( \frac{AW_Q(\hat{X}W_K)^{\top}}{\sqrt{d}} + \log M \right) \hat{X}W_V, \tag{1}$$

and injected back into the video stream via a gated residual update

$$\hat{X} \leftarrow \hat{X} + \left( M^{\top} \odot \sigma(AW_g) \right) A, \tag{2}$$

where $\sigma$ denotes a sigmoid gate. Here, $W_Q, W_K, W_V \in \mathbb{R}^{d \times d}$ are standard query, key, and value projection matrices, and $W_g \in \mathbb{R}^{d \times 1}$ is a gating projection. The enriched features $\hat{X}$, contextually grounded by the actions and objects described in the transcript, are then used for both TAS and LTA.

**Anticipation decoder.** Building upon these representations, we design a transformer-based parallel decoder adapted from Gong et al. (2022a) and Nawhal et al. (2022a), that operates on the fused encoder output, defined as $\tilde{F} \in \mathbb{R}^{T_{\text{obs}} \times d_{\text{TAS}}}$. This fused output is projected into the anticipation space and enriched with learnable positional embeddings, while a fixed set of queries $Q \in \mathbb{R}^{\mathcal{C}_{\text{LTA}} \times d_{\text{LTA}}}$ attends to $\tilde{F}$ through cross-attention to hypothesize possible future action segments. The resulting descriptors $S$ are decoded to $\mathcal{C} \leq \mathcal{C}_{\text{LTA}}$ action classes terminating when an <EOS> token is generated, treating anticipation as structured prediction. To further promote coherence, we apply a Conditional Random Field (CRF), inspired by TCCA (Maté & Dimiccoli, 2024), on top of the decoder outputs: while the transformer effectively captures global context, it may produce fragmented or inconsistent transitions. The CRF refines these predictions by modeling local dependencies between consecutive tokens, enforcing smooth and semantically valid action progressions across the anticipation timeline. Unlike prior approaches, our decoder leverages *weakly-supervised pseudo-labels* to guide training, making anticipation feasible without dense frame-level annotations.

## 3.2 TbLTA OBJECTIVE

Learning under transcript-level supervision poses a particularly challenging problem, as the model must jointly infer action boundaries and their durations in the observable part, and future continuations without access to frame-level annotations. In this context, the choice of loss functions becomes a central mechanism that enables effective training. The TbLTA framework is optimized through three complementary groups of losses: (i) *alignment-oriented losses*, which establish reliable alignments between transcripts and observed features; (ii) *segmentation-oriented losses* which ensure learning long-range temporal dependencies over the full video, and (iii) *anticipation-oriented losses*, which directly supervise the prediction of future sequences; The total objective is formulated as:

$$\mathcal{L} = \mathcal{L}_A + \mathcal{L}_{TAS} + \mathcal{L}_{LTA}, \tag{3}$$

where $\mathcal{L}_A$ aligns the transcripts, $\mathcal{L}_{TAS}$ makes transcripts actionable on the video and $\mathcal{L}_{LTA}$ enforces long-horizon structure on the future.

### 3.2.1 ALIGNMENT-ORIENTED LOSSES

We adopt an ATBA-style (Xu & Zheng, 2024) surrogate to obtain frame-wise pseudo-labels by aligning predictions to the observed transcript via dynamic programming over candidate boundaries. On top of these pseudo-labels, we apply a compact set of regularizers that proved crucial for stable training: (1) *Frame-wise cross-entropy* supervises per-frame predictions with ATBA pseudo-labels, (2) *Video-level multi-label classification* mitigates pseudo-label noise by supervising class presence at the clip level, and (3) *Global–local contrast* aligns class tokens with class-specific feature centroids to tighten semantics. We denote the weighted sum of these terms as $\mathcal{L}_{\text{atba}}$, and the total loss is defined as $\mathcal{L}_{\text{A}} = \gamma_1 \mathcal{L}_{\text{atba}}$. More details in the supplementary material.

### 3.2.2 SEGMENTATION-ORIENTED LOSSES

The Connectionist Temporal Classification (CTC) loss (Graves, 2012) was originally introduced for sequence labeling tasks where the alignment between input frames and target labels is unknown. Unlike hybrid approaches requiring Hidden Markov models, CTC enables end-to-end alignment by marginalizing over all possible frame-to-label paths that collapse to the transcript. This property makes it particularly suitable for weakly-supervised action learning, where only transcript-level annotations are available. By allowing flexible alignments between the transcript and the predicted action probabilities, CTC provides robust supervision for both the TAS head and the anticipation decoder, accommodating variable action durations without boundary annotations.

Formally, let $\mathcal{Y}$ denote the action transcript. We define the predicted action probabilities from the segmentation head as $\pi = [\pi_1, \ldots, \pi_{\alpha T}]$, with $\pi_t \in \mathcal{C} \cup \{\varnothing\}$, where $\varnothing$ denotes the blank label. The collapsing operator $\mathcal{B}(\pi)$ removes blanks and repeated labels to map a path $\pi$ into a valid transcript. The CTC objective that enforces transcript consistency is formulated as:

$$\mathcal{L}_{CTC} = -\log P(\mathcal{Y} \mid \pi), \qquad \text{where} \quad P(\mathcal{Y} \mid X) = \sum_{\pi \in \mathcal{B}^{-1}(\mathcal{Y})} \prod_{t=1}^{T} P(\pi_t \mid x_t). \qquad (4)$$

is the probability of generating transcript $\mathcal{Y}$ given a sequence of probability predictions. Here, $P(\pi_t \mid x_t)$ denotes the probability assigned to label $\pi_t$ at frame $t$. This alignment anchors the model by ensuring that the TAS heads remain consistent with the same transcript. As a result, the observed segment provides stable frame-level supervision, while the anticipated segment is constrained to follow the correct symbolic sequence. By marginalizing over all possible alignments, CTC removes the need for boundary annotations, prevents error propagation across modules, and becomes a supervisory signal that makes weakly-supervised long-term action anticipation feasible. We defined the $\mathcal{L}_{TAS} = \gamma_2 \mathcal{L}_{CTC}$.

### 3.2.3 ANTICIPATION-ORIENTED LOSSES

The total anticipation loss is a weighted combination of a global action sequence coherence loss ($\mathcal{L}_{\text{crf}}$) and a duration loss ($\mathcal{L}_{\text{dur}}$): $\mathcal{L}_{LTA} = \mathcal{L}_{\text{crf}} + \gamma_3 \mathcal{L}_{\text{dur}}$.

**Global action sequence coherence loss.** To promote temporally coherent forecasts, we place a linear-chain CRF on top of the anticipation decoder logits. Let the decoder output emission scores $Z \in \mathbb{R}^{T_{\text{pred}} \times |\mathcal{C}|}$, and let $\mathcal{Y}_{\text{LTA}}$ the target anticipate transcript. For a candidate sequence $c = (c_1, \ldots, c_{T_{\text{pred}}})$, the CRF score is

$$s(Z, c) = \sum_{t=1}^{T_{\text{pred}}} Z_{t,c_t} + \sum_{t=1}^{T_{\text{pred}}-1} M_{c_t, c_{t+1}}, \qquad (5)$$

where $M$ is a learnable transition matrix. The training objective is the negative log-likelihood of the ground-truth anticipation sequence:

$$\mathcal{L}_{\text{crf}} = -\log p(\mathcal{Y}_{\text{LTA}} \mid Z) = \log \sum_{c' \in \mathcal{C}^{T_{pred}}} e^{s(Z,c')} - s(Z, \mathcal{Y}_{\text{LTA}}). \qquad (6)$$

This loss enforces global sequence-level consistency and complements CTC, which ensures alignment at the frame level.

**Affinity-based duration loss**    Inspired by the affinity property of procedural videos, firstly introduced in Ding & Yao (2022), following which videos depicting the same activity share resembling action temporal portions, we propose a duration prediction head that is trained without any temporal ground truth. During training, we compute per-class duration estimates from the observed segments by counting the frequency of predicted labels from the segmentation head. These estimates are stored in a momentum-based buffer $\hat{d} \in \mathbb{R}^{|C|}$ that captures temporal priors in a self-supervised fashion. During inference, the decoder outputs the predicted class probabilities, and the class duration priors $\hat{d}$ are concatenated and passed to a regression head to obtain a per-segment predicted duration $\hat{\delta}_i$. The self-supervised duration loss is formulated as:

$$\mathcal{L}_{\text{dur}} = \frac{1}{T_{\text{pred}}} \sum_{i=1}^{T_{\text{pred}}} \left(\hat{\delta}_i - \hat{d}_{y_i}\right)^2, \tag{7}$$

where $\hat{\delta}_i$ is the per-segment predicted duration and the ground truth target is approximated by the class-wise prior $\hat{d}_{y_i}$. This term encourages consistent duration estimates aligned with implicitly learned temporal statistics.

## 4 EXPERIMENTS

### 4.1 EXPERIMENTAL SETUP

**Datasets.**    We evaluate our approach on two widely used benchmarks for long-term action anticipation. The *Breakfast* dataset (Kuehne et al., 2014) comprises 1,712 videos of 52 participants performing breakfast-related activities in diverse kitchen environments. Each video is annotated at two levels: 10 coarse activities and 48 fine-grained action classes. The average duration is 2.3 minutes, and the dataset exhibits a highly imbalanced action distribution (Ding & Yao, 2022). The *50Salads* dataset (Stein & McKenna, 2013) consists of 50 top-view RGB-D recordings of individuals preparing mixed salads, totaling over 4 hours of annotated footage and covering 17 fine-grained action classes. Compared to Breakfast, the videos are longer and typically contain around 20 action instances. The *EGTEA Gaze+* dataset (Li et al., 2018) comprises 28 hours of egocentric video with 10.3K annotated action instances, spanning 19 verbs, 51 nouns, and 106 distinct verb–noun action classes. For all datasets, we used pre-extracted 2048-dimensional I3D features (Carreira & Zisserman, 2018) as visual input $X$.

**Metrics.**    For *Breakfast* and *50Salads*, we report Mean over Classes (MoC) accuracy, which computes frame-wise accuracy per class and averages across classes (Abu Farha et al., 2018). Anticipation is evaluated at different horizons: the model observes an initial portion of the video ($\alpha = 20\%$ or $30\%$) and predicts the next $\beta = 10\%, 20\%, 30\%$, or $50\%$ of the sequence. Results are averaged over four standard splits for Breakfast and five for 50Salads. For *EGTEA Gaze+*, we adopt mean Average Precision (mAP) following the multi-label classification protocol of Nagarajan et al. (2020), where $\alpha \in 25\%, 50\%, 75\%$ of each video is observed and the remaining segment ($100\% - \alpha$) is predicted. We report mAP over all actions (All), low-shot (Rare), and many-shot (Freq) classes, restricting evaluation to verb prediction.

**Implementation details**    The overall architecture is illustrated in Fig. 2. The transformer encoder used for the Breakfast dataset employs 4 layers, a hidden dimension of 128, and 4 attention heads. For the *50Salads* dataset, we use a hidden dimension of 512, with 4 attention heads and 8 Transformer layers. For the text embeddings, we employ a simple pretrained model such as DistilBERT (Sanh et al., 2019). The LTA decoder employs a hidden dimension of 128 for Breakfast and 256 for 50Salads, using 2 and 3 Transformer layers, respectively. The CRF module adopts the same configuration as in (Maté & Dimiccoli, 2024). The number of learned queries is set to 8 for *Breakfast* and 20 for *50Salads*. For *EGTEA Gaze+*, we apply the same configuration as *50Salads*.

**Training and Inference.**    Since pseudo-labeling requires a reliable initialization, we adopt a progressive training scheme. The model is first pre-trained for 10 epochs using only the video-level classification loss $\mathcal{L}_{\text{vid}}$, which enhances pseudo-label quality and yields a stable starting point. We then run a short stage of 30 epochs with segmentation- and alignment-oriented losses ($\mathcal{L}_A + \mathcal{L}_{\text{TAS}}$) to refine temporal structure. Finally, end-to-end optimization is performed with the complete set of losses in Eq. 3. At the beginning of each stage, both optimizer state and learning-rate schedule are re-initialized to secure stable convergence. During training, the segmentation head processes the full

| Dataset | Category | Method | Obs 20% | | | | Obs 30% | | | | Avg. |
|---|---|---|---|---|---|---|---|---|---|---|---|
| | | | 10% | 20% | 30% | 50% | 10% | 20% | 30% | 50% | |
| 50Salads | Supervised | Cycle Cons. Abu Farha et al. (2020b) | 34.76 | 28.41 | 21.82 | 15.25 | 34.39 | 23.70 | 18.95 | 15.89 | 24.15 |
| | | FUTR Gong et al. (2022b) | **39.55** | 27.54 | 23.32 | 17.77 | 35.15 | 24.85 | 24.22 | 15.26 | 25.96 |
| | | ObjectPrompt Zhang et al. (2024c) | 37.40 | **28.90** | 24.20 | 18.10 | 28.00 | 24.00 | **24.30** | 19.30 | 25.53 |
| | | ActFusion Guo et al. (2024) | **39.55** | 28.60 | 23.61 | **19.90** | 42.80 | 27.11 | 23.48 | **22.07** | **28.39** |
| | Weakly supervised | WS-DA $^\dagger$ Zhang et al. (2021) | - | - | - | - | 21.30 | - | - | - | - |
| | | **Ours (TbLTA)** | 24.90 | 21.12 | 19.00 | 14.45 | 27.67 | 25.32 | 20.27 | 14.65 | 20.92 |
| | | **Ours (TbLTA)*** - *Mean* | 26.01 | 17.68 | 15.04 | 14.87 | 25.93 | 22.17 | 17.57 | 13.68 | 19.11 |
| | | **Ours (TbLTA)*** - *Top1* | 33.76 | 27.85 | 25.00 | 22.16 | 34.49 | 33.29 | 29.35 | 22.18 | 28.51 |
| Breakfast | Supervised | Cycle Cons. Abu Farha et al. (2020b) | 25.88 | 23.42 | 22.42 | 21.54 | 29.66 | 25.58 | 25.20 | 25.13 | |
| | | FUTR Gong et al. (2022b) | 27.70 | 24.55 | 22.83 | 22.04 | 32.37 | 29.88 | 27.49 | 25.87 | 26.59 |
| | | ActFusion Guo et al. (2024) | **28.25** | 25.52 | **24.66** | 23.25 | 35.79 | 31.76 | 29.64 | 28.78 | 28.45 |
| | Weakly supervised | WS-DA $^\dagger$ Zhang et al. (2021) | - | - | - | - | 15.65 | - | - | - | - |
| | | **Ours (TbLTA)** | 27.47 | **26.21** | 21.62 | 20.53 | **40.28** | **35.76** | **31.67** | **28.79** | **29.03** |
| | | **Ours (TbLTA)*** - *Mean* | 28.92 | 25.63 | 24.61 | 21.80 | 38.38 | 35.06 | 31.89 | 28.67 | 29.37 |
| | | **Ours (TbLTA)*** - *Top1* | 37.18 | 32.92 | 31.66 | 30.45 | 45.72 | 41.92 | 39.06 | 38.27 | 37.15 |

Table 1: Comparisons of action anticipation on the Breakfast (Kuehne et al., 2014) and 50Salads (Stein & McKenna, 2013) benchmarks using our proposed models. The highest accuracy under a deterministic framework is indicated in **bold**, and the second highest is underlined. The highest accuracy under a probabilistic framework is indicated in gray. WS-DA $^\dagger$ (Zhang et al., 2021) operates under a (semi-) weakly supervised setting, using frame-level labels only for the observed segment of the video during training. * means stochastic protocol.

video, while at inference, only a fraction is observed, following the protocol of Gong et al. (2024). We also report the stochastic protocol of Abu Farha & Gall (2019) in the supp. mat.

## 4.2 COMPARATIVE RESULTS

To assess the effectiveness of TbLTA, we follow the protocol established in previous work (Farha & Gall, 2019; Sener et al., 2020; Gong et al., 2022b; 2024): we report comparative results on 50Salads and Breakfast datasets in Tab. 1 and additionally on EGTEA in Tab. 2. TbLTA consistently surpasses prior (semi-) weakly-supervised baselines of (Zhang et al., 2021), establishing the first transcript-only benchmark for dense LTA. Remarkably, despite the absence of frame-level supervision, our deterministic model attains performance competitive with, and occasionally superior to, fully supervised approaches. On Breakfast, TbLTA exhibits a pronounced gain at 30% observation, outperforming all supervised baselines. This result highlights the ability of transcript-based supervision to capture the procedural regularities of activities. Performance on 50Salads paints a complementary picture. Here, long videos, denser action distributions, and frequent transitions yield weaker temporal regularities, amplifying the impact of imprecise temporal alignment in the absence of boundary annotations. In addition, we also report stochastic results, where TbLTA achieves substantially higher accuracy by capturing multiple plausible futures. This dual view, deterministic for reproducibility and stochastic for diversity, illustrates both the flexibility and the limits of our approach. Tab. 2 evaluates TbLTA on EGTEA, where supervised models retain a clear edge overall, but our method proves to be competitive on rare classes. This suggests that high-level semantic supervision from transcripts can mitigate data imbalance, even without dense frame labels. Taken together, these results highlight our central contribution: TbLTA is the first framework to make dense long-term anticipation feasible with transcript supervision alone. While fully-supervised models still dominate the paradigm, TbLTA demonstrates that transcript-based supervision is a promising paradigm for more scalable and language-informed LTA.

## 4.3 ABLATION STUDY

All ablations are conducted on both Breakfast and 50Salads, and we report results using the `Top-1 MoC` metric. For clarity, we adopt this choice Top-1 MoC for ablations as it provides a stable reference point.

**Effect of CTC loss.** Removing the CTC supervision consistently degrades the quality, as shown in 3. On 50Salads, the average accuracy drops by $\approx 0.6$ points, while on Breakfast, the decline is $\approx 0.8$ points. This confirms that CTC helps to stabilize pseudo-labels and prevent error accumulation across tasks. Without this alignment, pseudo-label noise propagates more strongly into the anticipation stage.

**Effect of Multimodal Cross-Attention.** We contrast our multimodal cross-attention with two baselines: (i) cross-att simplex, which embeds the transcript and applies a single, unconstrained cross-attention to video features, and (ii) w/o cross-att, which removes cross-modal conditioning. Results in Table 3 (TAS) and Table 4 (LTA) show a consistent hierarchy: *w/o cross-att < cross-att simplex <* **TbLTA**. On 50Salads, the average score decreases by $\approx 1.3$ points ($\approx 0.8$ with cross attention simplex), while on Breakfast, the drop reaches $\approx 5.7$ points ($\approx 3.8$ with cross attention

| Model | All | Freq | Rare |
|---|---|---|---|
| Timeception (Hussein et al., 2019) | 74.10 | 79.70 | 59.70 |
| Anticipatr (Nawhal et al., 2022b) | **76.80** | **83.30** | 55.10 |
| TbLTA | 65.37 | 73.46 | **60.11** |

Table 2: TbLTA results in EGTEA compared to supervised models.

| Dataset | Model | Obs 20% | | | | Obs 30% | | | | Avg. |
|---|---|---|---|---|---|---|---|---|---|---|
| | | 10% | 20% | 30% | 50% | 10% | 20% | 30% | 50% | |
| 50Salads | TbLTA | 33.8 | 27.9 | 25.0 | 22.1 | 34.5 | 33.3 | 29.4 | 22.2 | **28.5** |
| | w/o ctc loss | 32.3 | 29.3 | 25.2 | 21.0 | 34.2 | 32.5 | 29.1 | 19.7 | 27.9 |
| | w cross-att simplex | 31.1 | 26.8 | 24.3 | 21.8 | 33.6 | 33.1 | 29.3 | 21.7 | 27.7 |
| Breakfast | TbLTA | 37.2 | 33.0 | 31.7 | 30.5 | 45.7 | 41.9 | 39.1 | 38.3 | **37.2** |
| | w/o ctc loss | 36.0 | 31.7 | 31.0 | 30.1 | 44.2 | 41.4 | 38.8 | 37.6 | 36.4 |
| | w cross-att simplex | 30.4 | 26.7 | 27.0 | 27.9 | 42.7 | 38.7 | 37.1 | 36.7 | 33.4 |

Table 3: Ablation study on Alignment/TAS modules.

| Dataset | Model | Obs 20% | | | | Obs 30% | | | | Avg. |
|---|---|---|---|---|---|---|---|---|---|---|
| | | 10% | 20% | 30% | 50% | 10% | 20% | 30% | 50% | |
| 50Salads | TbLTA | 33.8 | 27.9 | 25.0 | 22.1 | 34.5 | 33.3 | 29.4 | 22.2 | **28.5** |
| | w/o duration | 31.1 | 29.2 | 24.7 | 20.2 | 38.2 | 33.8 | 29.2 | 19.8 | 28.3 |
| | w/o cross-att | 30.2 | 28.2 | 25.0 | 20.7 | 33.2 | 32.2 | 28.8 | 19.5 | 27.2 |
| | w/o CRF | 26.4 | 26.2 | 21.0 | 16.0 | 35.8 | 25.7 | 21.1 | 13.3 | 23.2 |
| Breakfast | TbLTA | 37.2 | 33.0 | 31.7 | 30.5 | 45.7 | 41.9 | 39.1 | 38.3 | **37.2** |
| | w/o duration | 34.1 | 30.4 | 27.6 | 22.5 | 46.6 | 41.7 | 37.1 | 30.8 | 33.9 |
| | w/o cross-att | 31.7 | 27.5 | 25.8 | 24.8 | 39.9 | 35.5 | 33.1 | 33.6 | 31.5 |
| | w/o CRF | 39.7 | 33.1 | 28.1 | 20.0 | 47.2 | 40.2 | 32.9 | 22.9 | 33.0 |

Table 4: Ablation study on LTA module on 50Salads and Breakfast datasets.

(a) Breakfast dataset.  (b) 50Salads dataset.

Figure 3: **Qualitative results.** We display the ground-truth (GT) and the results of TbLTA (Ours) on two datasets: (a) Breakfast and (b) 50Salads.

simplex). Overall, while the simplex variant provides some conditioning, it lacks the structural biases of our multimodal design—masking by transcript-derived neighborhoods and gated residual fusion—leading to inferior alignment and weaker long-horizon coherence.

**Effect of CRF loss.** The contribution of the CRF loss is particularly evident at longer horizons, as shown in Table 4. While short-term accuracy remains similar (even slightly higher on BF), its removal causes notable declines at longer horizons ($\approx 5.3$ on 50Salads, $\approx 4.1$ on Breakfast), underscoring its role in enforcing temporal coherence and stabilizing long-term forecasts.

**Effect of duration loss.** Table 4 shows that removing the duration loss reduces accuracy ($\approx 0.2$ on 50Salads, $\approx 3.3$ on Breakfast), indicating that it serves as a temporal regularizer that stabilizes long-horizon predictions by discouraging unrealistic segment durations. Since it is trained without temporal ground truth and relies on momentum-based class-wise priors, we use this term only as a weak duration prior rather than a precise per-instance predictor. Consistent with duration modeling in fully supervised LTA, its effect is most beneficial for actions with more concentrated duration statistics, while classes with high intra-class variability remain challenging.

### 4.4 QUALITATIVE RESULTS

Figures 3b and 3a illustrate representative qualitative results of our framework. The left part of each timeline (before the vertical dashed line) corresponds to the segmentation of the observed interval, while the right part (after the dashed line) shows the anticipated sequence of future actions. As can be seen, the model produces accurate and temporally coherent segmentations of the observed portion, and the degradation in prediction quality for the future interval remains relatively small. It also appears clear that an accurate prediction of action durations is still a challenge. More qualitative results are provided in the supp. mat.

## 5 CONCLUSION

We introduced TbLTA, the first framework for dense long-term action anticipation trained exclusively from transcripts, without requiring frame-level annotations. By combining temporal alignment to generate pseudo-labels with cross-modal attention to semantically ground video features, our model enables anticipation without dense supervision while preserving temporal action consistency over long horizons. Through extensive experiments on Breakfast, 50Salads, and EGTEA, TbLTA establishes the first transcript-based supervision baseline for LTA. Remarkably, despite the absence of dense labels, our model achieves results that are competitive with, and in certain settings even superior to, fully supervised methods. A major challenge that remains is to correctly estimate future durations, especially for unseen actions. Importantly, this work demonstrates that dense LTA does not needs to rely on exhaustive frame-level annotation, opening a new paradigm for scalable and language-informed anticipation in a weakly-supervised setting.

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
