# LONG-TERM ACTION ANTICIPATION VIA TRANSCRIPT BASED SUPERVISION

## A  APPENDIX

### A.1  SENSITIVE ANALYSIS.

Our method employs several loss functions whose relative contribution is controlled by three hyper-parameters: $\gamma_1$ weights the alignment-oriented losses, $\gamma_2$ is used for the segmentation module, and $\gamma_3$ for the anticipation module. We perform a sensitivity analysis over $\gamma_{1-3}$, and report the results on Breakfast dataset in Table 1. The grey-shaded rows correspond to the configurations adopted in all subsequent experiments.

Overall, performance is relatively stable over a broad range of values for each $\gamma_i$, with noticeable deviations only for more extreme settings (e.g., $\gamma_1 = 0.2$ or $\gamma_2 \geq 0.05$). This behaviour indicates that TbLTA is not overly sensitive to the precise choice of these hyperparameters. Within the stable region, the shaded configurations were selected because they offer the most favourable trade-off between **Mean** and **Top-1** accuracy. In all three cases, they achieve consistently high Mean scores while maintaining competitive Top-1 performance. By contrast, some settings with slightly higher Top-1 exhibit a clear degradation in Mean. For instance, $\gamma_2 = 0.1$ attains a Top-1 of 39.0 but only a Mean of 19.1, making them less desirable overall. Reporting both Mean and Top-1 in the table therefore makes the robustness range and this trade-off explicit, while justifying the final choice of the shaded hyperparameters.

### A.2  ADDITIONAL QUALITATIVE RESULTS

Figures 1 and 2 provide additional qualitative examples comparing segmentation and anticipation. The portion of each timeline preceding the vertical dashed line corresponds to the TAS output on the observed interval, while the portion after the dashed line illustrates the LTA of future actions. These examples highlight the ability of our model to produce temporally coherent segmentations of the observed video and to extend these predictions into plausible and semantically consistent future action sequences.

### A.3  ADDITIONAL RESULTS:

**Varying of CTC loss.**  We empirically found that enforcing transcript consistency via CTC on encoder frame-logits over the full video yields more stable alignments than applying CTC on concatenated TAS+LTA logits (see Table 3. This design isolates future uncertainty to the decoder and CRF, avoiding noisy global alignments while still transferring reliable boundary information to anticipation.

**Effect of TbLTA in TAS-only mode.**  We also evaluate TbLTA in a TAS-only setting ($\alpha = 1.0$, $\beta = 0.0$). The model obtains `MoF` = 54.5, `F1@10/25/50` = 37.0/29.8/18.9, and `Edit` = 32.2, which is competitive with weakly-supervised ATBA (`MoF` = 53.9) despite not being specialized for segmentation.

**Stochastic Evaluation Protocol**  For evaluation, we also closely follow prior work (Zatsarynna et al., 2025; 2024). Specifically, for each observed video snippet, we sample $S = 25$ predictions from our model. As evaluation metrics, we report:

- **Mean MoC:** the average Mean over Classes (MoC) accuracy across the $S$ generated samples.

| TbLTA | MoC | Varying | Obs 20% | | | | Obs 30% | | | | Avg. |
|---|---|---|---|---|---|---|---|---|---|---|---|
| | | | 10% | 20% | 30% | 50% | 10% | 20% | 30% | 50% | |
| $\gamma_1$ | Mean | 0.2 | 20.9 | 18.7 | 17.2 | 15.9 | 24.2 | 21.8 | 20.0 | 19.5 | 19.8 |
| | | 0.4 | 26.1 | 23.0 | 21.4 | 19.3 | 38.8 | 34.2 | 30.9 | 27.6 | 27.7 |
| | | 0.6 | 28.9 | 25.6 | 24.6 | 21.8 | 38.4 | 35.1 | 31.9 | 28.7 | **29.4** |
| | | 0.8 | 25.9 | 23.7 | 22.7 | 20.4 | 38.6 | 34.7 | 30.9 | 28.3 | _28.1_ |
| | | 1.0 | 25.6 | 23.5 | 22.5 | 19.9 | 38.2 | 34.2 | 30.4 | 27.3 | 27.7 |
| | Top-1 | 0.2 | 30.0 | 26.3 | 24.3 | 23.4 | 31.7 | 28.4 | 26.2 | 28.6 | 27.4 |
| | | 0.4 | 36.4 | 32.0 | 30.4 | 30.3 | 47.4 | 41.7 | 38.4 | 38.4 | _36.9_ |
| | | 0.6 | 37.2 | 32.9 | 31.7 | 30.5 | 45.7 | 41.9 | 39.1 | 38.3 | **37.2** |
| | | 0.8 | 34.4 | 31.4 | 30.2 | 29.5 | 45.3 | 40.6 | 37.5 | 37.9 | 35.9 |
| | | 1.0 | 34.8 | 32.2 | 31.1 | 29.0 | 45.1 | 40.6 | 37.4 | 36.2 | 35.8 |
| $\gamma_2$ | Mean | 0.001 | 25.4 | 23.2 | 21.8 | 19.7 | 37.2 | 33.7 | 30.6 | 27.8 | 27.4 |
| | | 0.005 | 27.3 | 24.3 | 22.5 | 20.5 | 38.5 | 34.6 | 31.0 | 27.8 | _28.3_ |
| | | 0.01 | 28.9 | 25.6 | 24.6 | 21.8 | 38.4 | 35.1 | 31.9 | 28.7 | **29.4** |
| | | 0.05 | 22.5 | 19.8 | 18.3 | 15.8 | 31.0 | 27.6 | 25.1 | 21.8 | 22.7 |
| | | 0.1 | 19.9 | 17.3 | 16.0 | 13.9 | 24.9 | 22.1 | 20.2 | 18.5 | 19.1 |
| | Top-1 | 0.001 | 34.4 | 31.0 | 29.8 | 29.6 | 45.5 | 41.1 | 38.6 | 39.1 | 36.1 |
| | | 0.005 | 37.4 | 33.5 | 31.9 | 31.0 | 47.7 | 42.6 | 39.4 | 38.8 | _37.8_ |
| | | 0.01 | 37.2 | 32.9 | 31.7 | 30.5 | 45.7 | 41.9 | 39.1 | 38.3 | 37.2 |
| | | 0.05 | 40.3 | 34.4 | 32.0 | 28.3 | 46.4 | 41.1 | 37.4 | 35.7 | 36.9 |
| | | 0.1 | 42.1 | 35.6 | 32.5 | 29.1 | 51.1 | 44.2 | 39.5 | 37.8 | **39.0** |
| $\gamma_3$ | Mean | 0.6 | 26.8 | 23.7 | 21.9 | 20.3 | 38.5 | 34.2 | 30.8 | 27.8 | 28.0 |
| | | 0.8 | 26.9 | 24.2 | 22.9 | 20.6 | 40.1 | 36.1 | 32.6 | 28.8 | _29.0_ |
| | | 1.0 | 28.9 | 25.6 | 24.6 | 21.8 | 38.4 | 35.1 | 31.9 | 28.7 | **29.4** |
| | | 1.2 | 27.7 | 24.9 | 23.3 | 20.6 | 39.6 | 35.6 | 32.0 | 28.6 | _29.0_ |
| | Top-1 | 0.6 | 35.4 | 31.4 | 29.5 | 29.7 | 46.4 | 41.0 | 37.6 | 37.3 | 36.0 |
| | | 0.8 | 36.7 | 33.0 | 32.1 | 30.8 | 48.6 | 43.9 | 40.7 | 39.9 | **38.2** |
| | | 1.0 | 37.2 | 32.9 | 31.7 | 30.5 | 45.7 | 41.9 | 39.1 | 38.3 | _37.2_ |
| | | 1.2 | 35.7 | 32.0 | 30.6 | 29.7 | 46.1 | 41.8 | 39.0 | 39.0 | 36.7 |

Table 1: Sensitive analysis for each $\gamma_{1-3}$ on the Breakfast dataset (Kuehne et al., 2014).

- **Top-1 MoC:** the MoC of the single best-matching sample among the $S$ candidates.

This protocol allows us to assess both the diversity (*Mean MoC*) and the best-case accuracy (*Top-1 MoC*) of the model under stochastic decoding.

### A.4  HYPERPARAMETERS

The full configuration for all datasets is reported in Table 2, which summarizes all hyperparameter choices used in our experiments.

### A.5  DETAILS ON LOSSES

**Temporal Alignment Losses.**  To train the segmentation branch under weak supervision, we adopt the loss formulation proposed in the Action Temporal Boundary Adjustment (ATBA) framework Xu & Zheng (2024). ATBA method generates frame-level pseudo-labels by aligning the model's predictions with transcript supervision, while promoting temporal coherence and boundary precision. ATBA begins by identifying a class-agnostic set of candidate boundaries, from which it selects the optimal $k_{\text{obs}} - 1$ transitions that best align with the observed transcript $\mathcal{Y}_{\text{obs}}$, using dynamic programming. This results in a frame-level sequence of pseudo-labels $\hat{Y} = [\hat{y}_1, \ldots, \hat{y}_{T_{\text{obs}}}]$ that serve as supervisory signals.

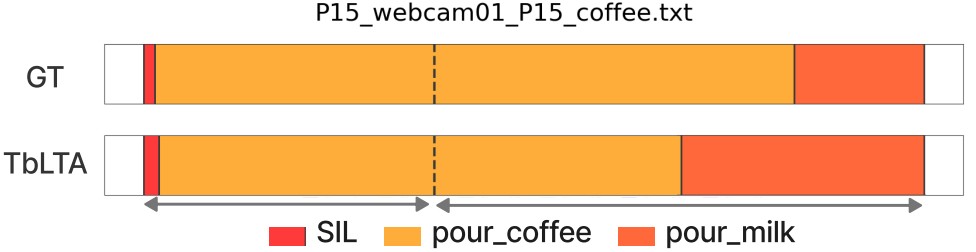

Figure 1: Qualitative results on Breakfast datasets.

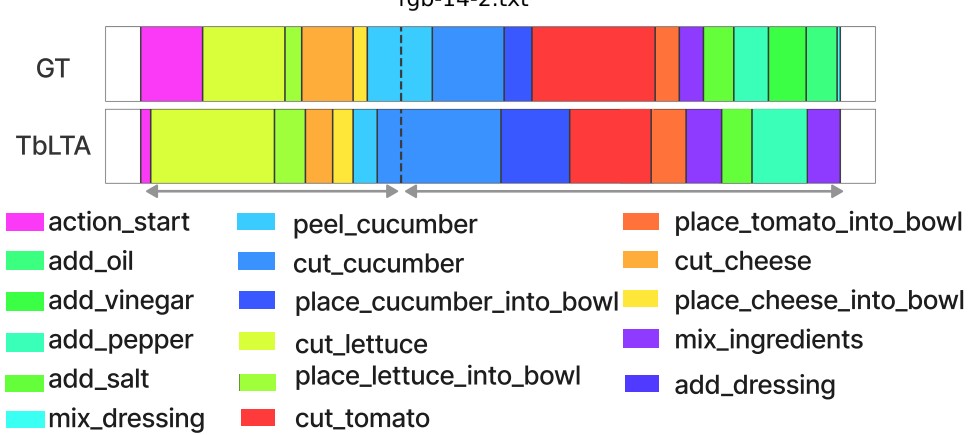

Figure 2: Qualitative results on 50Salads dataset.

| Hyperparameter | 50Salads/ EGTEA Gaze+ | Breakfast |
|---|---|---|
| Dataset classes ($\mathcal{C}$) | 19 | 48 |
| Batch size (bs) | 4 | 2 |
| Epochs (ne) | 80 | 80 |
| Learning rate (lr) | $5\times10^{-4}$ | $1\times10^{-4}$ |
| Weight decay (wd) | $3\times10^{-4}$ | $5\times10^{-5}$ |
| Feature dim ($f$) | 256 | 256 |
| Encoder input dim | 512 | 128 |
| Encoder layers | 8 | 4 |
| Decoder layers | 4 | 2 |
| Decoder hidden dim | 256 | 128 |
| Decoder heads | 4 | 8 |
| Decoder queries ($\mathcal{Q}$) | 20 | 8 |
| CRF weight | 1.0 | 0.1 |
| Dropout | 0.5 | – |
| Top-$k$ (CRF) | 25 | 25 |
| Text encoder | DistilBERT | DistilBERT |
| $\gamma_1$ | 0.6 | 0.6 |
| $\gamma_2$ | 0.01 | 0.01 |
| $\gamma_3$ | 1.0 | 1.0 |

Table 2: Hyperparameter configuration on all datasets.

Given the fused representation $\tilde{F} \in \mathbb{R}^{(|C|+T_{\text{obs}})\times d_{TAS}}$, we estimate action probabilities through:

$$\xi_{seg} = \sigma(W_{seg}\tilde{F} + \epsilon_{seg}) \in \mathbb{R}^{(|C|+T_{\text{obs}})\times |C|} \tag{1}$$

where $W_{seg}$ and $\epsilon_{seg}$ are learnable parameters and $\sigma(\cdot)$ denotes the sigmoid activation. We define the following loss components:

| TbLTA | Model | Obs 20% | | | | Obs 30% | | | | Avg. |
|---|---|---|---|---|---|---|---|---|---|---|
| | | 10% | 20% | 30% | 50% | 10% | 20% | 30% | 50% | |
| 50Salads | ctc loss (obs) | 33.8 | 27.9 | 25.0 | 22.1 | 34.5 | 33.3 | 29.4 | 22.2 | **28.5** |
| | ctc loss (full video) | 26.6 | 26.8 | 24.5 | 21.9 | 33.4 | 34.2 | 28.7 | 22.2 | 27.3 |
| Breakfast | ctc loss (obs) | 37.2 | 33.0 | 31.7 | 30.5 | 45.7 | 41.9 | 39.1 | 38.3 | **37.2** |
| | ctc loss (full video) | 35.4 | 31.1 | 30.0 | 28.3 | 44.5 | 41.1 | 38.3 | 33.1 | 35.2 |

Table 3: Ablation study on CTC loss.

- *Frame-wise Cross-Entropy Loss:*
  Let $P_{\text{frames}} \subset \xi_{\text{seg}}$ denote the last $T_{\text{obs}}$ rows, corresponding to the frame-level predictions. We supervise them using pseudo-labels:

$$\mathcal{L}_{\text{frames}} = -\frac{1}{T_{\text{obs}}} \sum_{t=1}^{T_{\text{obs}}} \log P_{\text{frames}_{t, \hat{y}_t}} \tag{2}$$

- *Video-level Binary Classification Loss:*
  To address noise in pseudo-labels, we follow Xu & Zheng (2024) and add a video-level multi-label classification loss:

$$\mathcal{L}_{\text{vid}} = -\frac{1}{|C|} \sum_{c=1}^{|C|} \left[ y_c^{\text{vid}} \log \xi_{\text{seg}_c} + (1 - y_c^{\text{vid}}) \log(1 - \xi_{\text{seg}_c}) \right] \tag{3}$$

  where $\xi_{\text{seg}_c}$ is the action occurrence probability for each class $c \in C$, $y_c^{\text{vid}} = 1$ if class $c$ is present in $\mathcal{Y}_{\text{obs}}$, and 0 otherwise.

- *Global-Local Contrastive Loss:*
  Class tokens $E' = [e'_1, \ldots, e'_{|C|}]$ encode global action semantics. For each class $c$, we compute a centroid $\bar{x}_c$ by averaging the frame features corresponding to class $c$ in $\hat{Y}$. A contrastive loss (He et al., 2020) then aligns local and global features:

$$\mathcal{L}_{\text{glc}} = -\frac{1}{|\text{Set}(\mathcal{Y}_{\text{obs}})|} \sum_{c \in \text{Set}(\mathcal{Y}_{\text{obs}})} \log \frac{\exp\left(\langle \bar{x}_c, e'_c \rangle / \tau\right)}{\sum_{c'=1}^{|C|} \exp\left(\langle \bar{x}_c, e'_{c'} \rangle / \tau\right)}, \tag{4}$$

  where $\langle \cdot, \cdot \rangle$ denotes cosine similarity and $\tau$ is a temperature hyperparameter.
  This loss complements frame-level supervision by encouraging the decoder to produce semantically meaningful and temporally coherent segments.

The final segmentation loss is a weighted combination of all terms:

$$\mathcal{L}_{atba} = \beta_1 \mathcal{L}_{\text{frames}} + \beta_2 \mathcal{L}_{\text{vid}} + \beta_3 \mathcal{L}_{\text{glc}} \tag{5}$$

where $\beta_1, \beta_2 = 0.5$ and $\beta_3 = 0.1$, following the ATBA paper.

**LTA Losses.** To supervise the LTA branch, we rely on the temporal consistency regularization strategy introduced by TCCA (Maté & Dimiccoli, 2024). The output sequence is predicted by a decoder and refined through a CRF. We define four loss components for this stage.

The primary objective is a CRF sequence loss $\mathcal{L}_{\text{crf}}$, which maximizes the log-likelihood of the predicted future label sequence under a CRF model. This ensures structural consistency in the anticipated sequence.

To promote contextual regularization, following BACR (Bi-Directional Action Context Regularizer), we apply two KL divergence terms. The first,

$$\mathcal{L}_{\text{next}} = D_{KL}(p_{\text{fut-cur}} \parallel p_{\text{fut-next}}), \tag{6}$$

aligns the current action distribution with that of the subsequent predicted action. The second,

$$\mathcal{L}_{\text{prev}} = D_{KL}(p_{\text{fut-cur}} \parallel p_{\text{fut-prev}}), \tag{7}$$

performs a similar alignment with the preceding action. These losses serve to improve transition coherence and reduce label jittering in the anticipated future.

The total CRF loss is then:

$$\mathcal{L}_{CRF} = \mathcal{L}_{\text{crf}} + \mathcal{L}_{\text{next}} + \mathcal{L}_{\text{prev}} \tag{8}$$

**Duration loss.** During training, we compute per-class duration estimates from the observed segments by counting the frequency of predicted labels from the segmentation head. These estimates are stored in a momentum-based buffer $\hat{d} \in \mathbb{R}^{|C|}$, updated as:

$$\hat{d}^{(t)} = w_b \cdot \hat{d}^{(t-1)} + (1 - w_b) \cdot d^{\text{batch}}, \tag{9}$$

where $w_b \in [0, 1]$ is a momentum balancing weight, and $d^{\text{batch}}$ is the mean observed duration for each class in the current batch. This running average captures temporal priors in a self-supervised fashion, even in the absence of true duration annotations. During inference, the decoder output $S_{LTA}$, the predicted class probabilities $\xi_{\text{fut-cur}}$, and the class duration priors $\hat{d}$ are concatenated and passed to a regression head to obtain a per-segment predicted duration:

$$\hat{\delta}_i = W_{\text{dur}} \cdot \left[ S_{LTA_i}, \xi_{\text{fut-cur}}^i, \hat{d} \right] + \epsilon_{\text{dur}}, \quad i = 1, \ldots, n_q^{LTA} \tag{10}$$

where $W_{\text{dur}}$ and $\epsilon_{\text{dur}}$ are learnable parameters of the duration prediction head, and $\hat{\delta}_i$ is the predicted normalized duration for segment $i$. The self-supervised duration loss is formulated as:

$$\mathcal{L}_{\text{dur}} = \frac{1}{T_{\text{pred}}} \sum_{i=1}^{T_{\text{pred}}} \left( \hat{\delta}_i - \hat{d}_{y_i} \right)^2, \tag{11}$$

where $\hat{\delta}_i$ is the per-segment predicted duration and the ground truth target is approximated by the class-wise prior $\hat{d}_{y_i}$.

## A.6 CRF DECODING.

In our framework, the Conditional Random Field (CRF) can produce multiple future action hypotheses. By default, setting `top_k` in the decoder returns $K$ candidate sequences, which may be sampled or enumerated depending on the implementation. A naïve strategy is to retain the first hypothesis ($k = 0$), implicitly assuming that the output is sorted by probability. However, this assumption does not always hold: in practice, the returned set can mix high- and low-probability sequences in arbitrary order.

To remove this ambiguity and ensure reproducibility, we adopt a deterministic decoding procedure. During evaluation, we generate $K = 25$ candidate futures from the CRF, compute the CRF score of each candidate (emissions plus transition potentials), and retain the most probable one. This guarantees determinism and ensures that reported metrics reflect the model's own highest-probability prediction rather than oracle or sampling artifacts.

## A.7 AI ASSISTANCE

We would like to note that large language models (ChatGPT, Gemini) were used to polishing the writing of this work.