# OpenReview forum: "Long-Term Action Anticipation via Transcript-based Supervision"
_ICLR.cc/2026/Conference — ICLR 2026 Conference Withdrawn Submission_

### Official Review · Reviewer_i3xn · 2025-10-27

**Soundness:** 2
**Presentation:** 1
**Contribution:** 2
**Rating:** 2
**Confidence:** 3

**Summary:**

This paper presents TbLTA, a transcript-based weakly supervised framework for long-term action anticipation (LTA). The method eliminates dense frame-level annotations by aligning ordered transcripts with video frames using a temporal alignment module (ATBA) to generate pseudo-labels. It combines multiple objectives—alignment, segmentation (CTC), and anticipation (CRF and duration losses)—and leverages cross-modal attention between transcript embeddings and video features for contextual grounding. TbLTA achieves good performance on Breakfast, 50Salads, and EGTEA benchmarks, approaching supervised baselines despite using action-order supervision only.

**Strengths:**

S1. First weakly supervised LTA framework: The paper introduces the first transcript-based approach for dense long-term anticipation, addressing scalability issues of fully supervised LTA.

S2. Competitive results: TbLTA achieves strong performance without frame-level annotations, particularly on Breakfast, comparable to fully supervised methods.

**Weaknesses:**

W1. Use of transcripts at inference: The method assumes transcript availability during inference (Sec. 3.1, L196–200), which raises concerns about practicality since ground-truth transcripts are typically unavailable at inference time in real-world settings. This assumption’s realism is questionable and should be clearly justified or relaxed.

W2. Limited technical novelty: Most components (ATBA, CTC, CRF, duration loss) are adopted from prior works and combined as weakly supervised LTA rather than introducing fundamentally new modeling ideas.

W3. Overly complex design: The framework introduces eight loss terms (3, 1, 4 for alignment, TAS, and LTA), multiple heuristics (e.g., affinity prior in duration loss), and multi-stage training, raising concerns about reproducibility, stability, and generalization. Sensitivity analysis on hyperparameters (λ₁–₃) is missing.

W4. Missing validation of multi-stage training: The paper does not empirically verify whether the progressive training schedule contributes to the final performance.

**Questions:**

Q1. The term “transcript” may be confused with ASR transcripts. Would a different term improve clarity?

Q2. Regarding W1, How is cross-attention with transcript embeddings computed during inference, when no ground-truth transcript embedding is available?

---

> ### Author Response · Authors · 2025-11-25
> **Answer to Reviewer i3xn (Part 1)**
>
> > W1. Use of transcripts at inference:
> We sincerely thank the reviewer for pointing out the inconsistency between the figure and the textual description regarding the use of transcripts at inference time. The confusion was caused by an incorrect sentence in Section 3.1 (line 196 in the original submission) that mistakenly suggested the use of transcripts at test time. We have updated the paper version to remove the wrong statement
>
> “2) At inference time, given only the observed features $X_{obs}$​ and the video transcript Y,”
> and replace it with
>
> “2) At inference time, given only the observed features $X_{obs}$ and the learned class tokens E.”
>
> TbLTA does **not** require ground-truth transcripts at inference time, as Fig. 2 is intended to show. In the revised version of the figure, we believe the training flow is now presented more clearly, since we explicitly separate the training and inference pipelines to reduce visual complexity and avoid misunderstandings (with training and inference flows depicted using distinct arrow styles/colors). Transcripts are used **only during training** as a weak supervision signal to
>
> (1) generate frame-level pseudo-labels via the temporal alignment module,
>
> (2) enforce sequence consistency through the CTC loss, and (3) provide semantic guidance via cross-modal attention. At test time, the model observes only the visual features of the partial video $X_{obs}$​ and the class tokens E learned during training; the anticipation decoder operates on their concatenation $[E∥Xobs]$ and its learned queries to produce future action sequences and durations, without any access to ground-truth transcripts.
>
> To further improve clarity on this important point, we explicitly reiterate in the abstract, introduction, Sec. 3.1, and in the captions of Fig. 1 and Fig. 2 that transcripts are used only during training and are not required at inference. We sincerely apologize for the confusion our incorrect wording may have caused.
>
> > W3. and W4. Overly complex design and multistage training.
>
> We thank the reviewer for these thoughtful comments regarding the complexity of our framework, the multi-stage training schedule, and the missing sensitivity analysis on the loss weights.
>
> We wish to clarify that these 8 terms are not 8 unique loss types, but rather a standard multi-task learning (MTL) framework supervising our three core components:
>
> **1. Alignment:** [e.g., $L_{atba}$]
>
> **2. TAS:** [e.g., $L_{CTC}$]
>
> **3. LTA:** [e.g., $L_{CRF}$, $L_{duration}$, etc.]
>
> This breakdown is essential for our progressive training and to provide precise supervision to each sub-task. We have already shown in our main paper's ablation study (Tables 3 and 4) that these components all contribute to the final performance.
>
> For the same reason, we adopt a progressive, multi-stage training schedule. In our setting, a reasonable initialization of earlier modules is crucial for the stability and effectiveness of the subsequent ones—for instance, the quality of the LTA predictions heavily depends on the TAS module it receives inputs from. To validate this design choice, we trained a variant where all components are optimized jointly from scratch, without the progressive schedule. As reported in the common results (Table 2) in our general response to all reviewers, this “single-stage” variant exhibits a substantial performance drop compared to our progressive training, confirming that multi-stage training is not merely an implementation detail but an important factor for achieving strong anticipation performance.
>
> We believe the reviewer is referring to the loss weights $\gamma_ {1-3}$. The sensitivity analysis on the gamma is a valid point. We are currently running an exhaustive sensitivity analysis around these weights and will upload the corresponding results and settings to the revised version **before the rebuttal deadline**, to further support reproducibility and clarify the impact of these hyperparameters.
>
> **Regarding the reproducibility of our work, we have explicitly stated in the revised version of the paper that we will make the code available upon acceptance.**

---

> ### Author Response · Authors · 2025-11-25
> **Answer to Reviewer i3xn (Part 2)**
>
> > Q1. The term “transcript” may be confused with ASR transcripts. Would a different term improve clarity?
>
> We thank the reviewer for this comment. The term transcript is widely used in the literature about action segmentation, and that’s why we adopted it for the closely related task of dense long-term anticipation. To avoid misunderstanding,  we added a short sentence in the revised version (Lines 157-161).
>
> > Q2. How is cross-attention with transcript embeddings computed during inference, when no ground-truth transcript embedding is available?
>
> Please see the answer to W1. In our current implementation, the cross-modal attention block is used **only during training** as a regularizer that injects transcript semantics into the video representation. The text embeddings are computed from the vocabulary of action labels (class names) and are combined with pseudo-labels to build local attention masks, but this happens exclusively in the training phase. At inference time, the anticipation decoder consumes the video features that have already internalized this semantic guidance; no transcript (or ASR-style transcript) is needed. We’ve updated Sec. 3.1 and the caption of Fig. 2 to make this training-only use of transcripts and cross-attention explicit.
>
>
> **Closing Note:** We thank the reviewer for the detailed feedback on the use of transcripts at inference, the perceived complexity of the design, and the need to validate the multi-stage training scheme. Your comments prompted several key corrections and new ablations that we believe significantly clarify the method. We sincerely hope these points will allow you to revisit your score. We remain fully open to any further questions, suggestions, or discussion.

---

> > ### Author Response · Authors · 2025-12-02
> > **Additional sensitivity analysis on $\gamma_{1-3}$**
> >
> > We thank the reviewer for this valuable suggestion. Following your suggestion (W3), **we have updated the supplementary material** by adding a dedicated **sensitivity study** of the three loss-weight hyperparameters (**Appendix A.1, Table 1**). We systematically vary each $\gamma_{i}$ while keeping the others fixed and report both Mean and Top-1 MoC on Breakfast. The results show that TbLTA is stable over a broad range of values, with noticeable degradation only for more extreme settings (e.g., very low $\gamma_{1}$ or $\gamma_{2}$ ≥ 0.05). Within the stable region, the configuration used in the main paper ($\gamma_{1}$= 0.6, $\gamma_{2}$ = 0.01, $\gamma_{3}$ = 1.0) provides one of the best trade-offs, achieving consistently high Mean while maintaining competitive Top-1. By contrast, some configurations with slightly higher Top-1 (e.g., $\gamma_{2}$ = 0.1) exhibit a clear drop in Mean, making them less desirable overall.
> >
> > **We sincerely thank you for your time and constructive feedback.** We kindly invite you to also review our common response addressed to all reviewers, where we summarize the main changes and additional experiments. **We hope that our clarifications and new analyses satisfactorily address your concerns and that you may consider revisiting your overall score for the paper.**

---

### Official Review · Reviewer_6poY · 2025-10-30

**Soundness:** 3
**Presentation:** 2
**Contribution:** 2
**Rating:** 4
**Confidence:** 5

**Summary:**

• The author proposed TbLTA as the first weakly-supervised approach for dense Long-Term Action Anticipation (LTA), utilizing only video transcripts that ordered action lists without timing to overcome the scalability issues associated with costly dense frame-level annotations.

• The model uses an encoder-decoder architecture that leverages a weakly-supervised temporal alignment module to generate dense pseudo-labels, providing crucial frame-level supervision for training the segmentation head and anticipation decoder.

• Transcripts are further exploited as semantic context to enrich video features via a local cross-modal attention mechanism and supply robust global supervision to the action segmentation predictions using a Connectionist Temporal Classification (CTC) loss.

• TbLTA establishes the pioneer transcript-only supervision baseline for LTA on benchmarks like Breakfast and 50Salads, achieving performance competitive with fully supervised methods.

**Strengths:**

- The work introduces a weakly-supervised framework for the dense LTA task, training exclusively on ordered action lists without temporal information. This eliminates the need for expensive frame-level boundary annotations for LTA benchmarks.


- The weakly supervised model achieves performance that is competitive with fully supervised methods on the Breakfast dataset, showing its ability to capture procedural regularities.


- The architecture incorporates transcripts for various supervision:
(1) Temporal alignment module generates frame-level pseudo-labels for dense supervision.
(2) CTC loss enforces sequence consistency, which is crucial for stabilizing pseudo-labels and preventing error propagation.
(3) Cross-modal attention layer leverages transcript semantics to contextually enrich video features.

**Weaknesses:**

1. **Justification of Weak Supervision:** The paper claims to address a major limitation, stating that Long-Term Action Anticipation (LTA) has been tackled exclusively in a fully supervised manner and that highly granular labeling is costly and difficult to scale. However, the relevance of this weakly-supervised setting is also tackled by relevant contemporary work, PALM [1], which already addresses action anticipation without time annotation, using a different form of weak supervision, a few-shot setting. The paper's claim that its approach is the "first fully weakly-supervised framework for dense LTA" must be rigorously justified against such contemporary models, especially if they operate in a shot setting without explicit time boundaries, which aligns to reduce the annotation burden.


2. **Exclusion of LTA Benchmarks:** The evaluation is limited to established procedural activity datasets: Breakfast, 50Salads, and EGTEA. The paper does not include experiments on the Ego4D dataset, which is one of the most well-known LTA datasets. Since Ego4D often involves complex, first-person (egocentric) interactions and activities, excluding it raises questions about the framework's scalability and generalizability to diverse, modern LTA settings, especially those relying on egocentric video where action boundaries might be less distinct than in procedural cooking videos.

3. **Lack of Conceptual Advancement and Incremental Methodology:** The methodology appears to be a robust assembly of existing techniques, potentially lacking fundamental new conceptual insights:

    (a) Architecture: The model is a transformer-based encoder-decoder architecture, similar to paradigms found in prior supervised LTA work like FUTR and ANTICIPATR.

    (b) Alignment: The core weakly-supervised component, the temporal alignment module, adopts the ATBA module proposed in Xu & Zheng (2024).

    (c) Losses: It utilizes well-established sequence-to-sequence learning objectives like the Connectionist Temporal Classification (CTC) loss (pioneered for segmentation by Huang et al. (2016)) and a Conditional Random Field (CRF) loss (adapted from Maté & Dimiccoli (2024)).



[1] Kim, Sanghwan, et al. "Palm: Predicting actions through language models." European Conference on Computer Vision. Cham: Springer Nature Switzerland, 2024.

**Questions:**

1. Please address the questions raised in the weakness section.


2. The paper explicitly identifies that an accurate prediction of action durations is still a challenge, especially for unseen actions. Given that the affinity-based duration loss **($L_{dur}$)** is trained without temporal ground truth and relies on momentum-based class-wise prior estimates derived from predicted labels, how robust is this self-supervised mechanism when the actual duration of an anticipated action significantly deviates from the implicitly learned temporal statistics captured by the buffer?


3. TbLTA’s success relies on generating frame-level pseudo-labels via a temporal alignment module. The CTC loss is critical as it stabilizes these pseudo-labels and prevents error accumulation. Since performance on datasets like 50Salads suffers because denser action distributions and frequent transitions amplify the impact of imprecise temporal alignment, how effectively does the CTC loss mitigate the noise and instability inherent in the generated pseudo-labels near action boundaries in dense video sequences?

---

> ### Author Response · Authors · 2025-11-25
> **Answer to Reviewer 6poY (Part 1)**
>
> > W1. and Q1. Justification of weak supervision.
>
> We thank the reviewer for raising this important point and for drawing our attention to PALM [1]. We agree that it should have been explicitly discussed in our related work, and we have added it in the revised version.
>
> At the same time, we would like to clarify that the supervision regime and task setup in PALM are substantially different from ours. PALM aims at predicting symbolic sequences of future actions (more precisely, K stochastic sequences of fixed length) evaluated by edit distance, rather than performing dense, frame-level long-term anticipation as in our work (i.e., it does not model action durations explicitly). Moreover, as stated in PALM Sec. 3.1, the method operates on input videos that are already segmented into P action units whose temporal boundaries are provided. In other words, PALM relies on fully supervised segment-level annotations and does not need to infer action boundaries or durations at training time.
>
> In contrast, our framework works in a boundary-free regime where no temporal annotations are available: the transcript provides only an ordered list of actions, with no information about start/end times, correspondence to observed vs. unobserved portions of the video, or per-action durations. TbLTA must therefore jointly infer segmentation, durations, and future actions directly from raw untrimmed video. To the best of our knowledge, no prior work addresses dense LTA under this transcript-only, fully weakly supervised temporal setting.
> To further avoid confusion, in the revised manuscript, we (i) explicitly position PALM in the LTA related-work section as a language-based, **non-dense anticipation method**, and (ii) are happy to **revise the title** to make our focus on dense LTA explicit (e.g., “Long-Term Dense Action Anticipation via Transcript-based Supervision”), if the reviewers consider this clearer.
>
>
> > W2. Lack of comparison to EGO4D:
>
> We appreciate the reviewer’s suggestion to include experiments on Ego4D and fully agree that it is a highly relevant benchmark for long-term anticipation. However, we would like to kindly clarify that, to the best of our knowledge, Ego4D has so far been used to evaluate non-dense LTA settings (as in PALM, discussed above), rather than dense, frame-level LTA with explicit duration modeling, which is the focus of our work.
>
> In contrast, our experimental protocol follows prior work on **dense LTA**, for which the established benchmarks are Breakfast, 50Salads, and EGTEA. We use all datasets that have been employed for dense LTA in the literature [1, 2]. For example, even a recent NeurIPS 2024 paper on dense LTA [2] reports dense anticipation results only on Breakfast and 50Salads, while using EGTEA exclusively for segmentation. In this sense, our evaluation setup is aligned with the current practice for dense LTA.
> Extending TbLTA to Ego4D (where both the task definition and evaluation protocol differ from dense LTA) would require additional design and adaptation steps that go beyond the scope of this work, but we agree it is an important and interesting direction for future research.
>
> [1] Gong, D., Lee, J., Kim, M., Ha, S. J., & Cho, M. (2022). Future transformer for long-term action anticipation. In Proceedings of the IEEE/CVF Conference on Computer Vision and Pattern Recognition (pp. 3052-3061).
>
> [2] Gong, D., Kwak, S., & Cho, M. (2024). Actfusion: a unified diffusion model for action segmentation and anticipation. Advances in Neural Information Processing Systems, 37, 89913-89942.

---

> ### Author Response · Authors · 2025-11-25
> **Answer to Reviewer 6poY (Part 2)**
>
> > Q2. The paper explicitly identifies that an accurate prediction of action durations is still a challenge, especially for unseen actions. Given that the affinity-based duration loss is trained without temporal ground truth and relies on momentum-based class-wise prior estimates derived from predicted labels, how robust is this self-supervised mechanism when the actual duration of an anticipated action significantly deviates from the implicitly learned temporal statistics captured by the buffer?
>
> We thank the reviewer for this insightful question. We agree that accurate duration prediction, especially for unseen or highly variable actions, remains challenging. Our affinity-based duration loss is indeed driven purely by momentum-based class-wise priors estimated from predictions, without temporal ground truth. In that sense, it behaves similarly to duration modeling in fully supervised methods, where models also tend to approximate average durations in the ground truth and struggle to capture large intra-class variability.
>
> In TbLTA, this term is therefore used only as a soft prior, not as a strict constraint: it gently encourages predicted durations to be consistent with the learned statistics. Empirically, we observe modest but consistent gains, mainly as smoother and more stable predictions for classes with reasonably concentrated durations.
>
> In the revised version L458-463, we clarified this point in the text, explicitly framing the duration term as a weak regularizer rather than a precise duration predictor, and we highlight more clearly that robust duration modeling is an open challenge shared with fully supervised LTA.
>
>
> > Q3. TbLTA’s success relies on generating frame-level pseudo-labels via a temporal alignment module. The CTC loss is critical as it stabilizes these pseudo-labels and prevents error accumulation. Since performance on datasets like 50Salads suffers because denser action distributions and frequent transitions amplify the impact of imprecise temporal alignment, how effectively does the CTC loss mitigate the noise and instability inherent in the generated pseudo-labels near action boundaries in dense video sequences?
>
> We thank the reviewer for this precise and important question.
> In TbLTA, the CTC loss complements the ATBA-based pseudo-label supervision by operating directly at the sequence level, without depending on frame-level pseudo-labels. This helps in dense datasets such as 50Salads in two main ways:
>
> **(1) Global sequence consistency.** CTC constrains the predicted sequence to match the transcript with minimal insertions, deletions, or re-orderings, discouraging spurious short segments and label “flickering” around boundaries, which are typical artifacts of noisy alignment in dense scenarios.
>
> **(2) Soft regularization of boundaries and durations.** Although CTC does not supervise exact frame boundaries, it encourages contiguous segments that explain the transcript coherently, which **indirectly stabilizes durations** when pseudo-labels near boundaries are fragmented or slightly misaligned.
>
> **Closing Note:** We thank the reviewer for the thorough and insightful review, as well as for highlighting the importance of our setting and for raising questions about PALM, duration modeling, and the impact of CTC in dense scenarios. The additional experiments and clarifications in the revised version are directly motivated by these points. We sincerely hope these points will allow you to revisit your score. We remain fully open to any further questions, suggestions, or discussion.

---

### Official Review · Reviewer_kytL · 2025-11-01

**Soundness:** 2
**Presentation:** 2
**Contribution:** 2
**Rating:** 2
**Confidence:** 3

**Summary:**

The paper tackles dense long-term action anticipation (LTA) without frame-level annotations by introducing TbLTA, a transcript-only, weakly supervised framework. To leverage the information from the transcript, the model introduces different loss objectives to align the provided transcript and visual features, which further improves the temporal action segmentation and future action anticipation.

**Strengths:**

1. The performance is better than existing work in 50Salads and Breakfast datasets.
2. The ablation study proves the effectiveness of the introduced loss and modules.

**Weaknesses:**

1. The drawn figure is bad. It is not clear for review.
2. Though the performance is better, it is not well motivated to have additional transcript information. The purpose of the WS-TAL/LTA task is mainly solving the insufficient label for action localization/anticipation. The introduction of transcript will take more annotation efforts and go against the original purpose.
3. The novelty of the proposed method seems very limited; most modules or methods are mainly from previous work, and this paper is just introducing transcript-based supervision for the LTA task, with no additional interesting module proposed.
4. It is a conflict between Fig. 2 and the content in line 196; the figure shows there is no transcript for inference, while it has one in the content.

**Questions:**

1. Make the figure clearer and readable.
2. Show strong motivation for the introduced transcript.
3. The novelty of the proposed method is very limited.
4. Consistent between the figure and text content.

---

> ### Author Response · Authors · 2025-11-25
> **Answer to Reviewer kytL**
>
> > W1. and Q1. Make the figure clearer and readable.
>
> Thank you for pointing out that the figure could be clearer and easier to read. We agree with this concern and have substantially revised the visual presentation of our architecture in the new version of the paper.
> In particular, we have redesigned Fig. 2 to more clearly indicate which modules are involved during training and which are not.
> Additionally, we have slightly modified Fig. 1, as well as the captions of both Fig. 1 and Fig. 2, to improve consistency and readability and to better highlight the roles of the main components of our model: (1) the temporal alignment module, (2) the multimodal cross-attention module, and (3) the global loss.
>
> **Besides the improvements we are proposing here, we are open to following any other concrete suggestion of improvement the reviewer may have about figures.**
>
>
> > W2. and Q2. Is not well motivated to have additional transcript information.
>
> Video transcript is the **only information** we use for training, so it is not an addition but rather a subtraction to dense frame level annotations used by the fully supervised methods we compare to. We indeed drastically reduce annotation effort by using only a sequence of class labels without boundary annotation (transcript), instead of dense annotations (one label per frame).
> We understand that our wrong writing on line 196 has led to a misunderstanding of this crucial point, and we apologize for that. We hope the reviewer will reconsider our motivation, novelty, and results in light of these important clarifications.
>
> > W4. and Q4. Consistency between the figure and text content.
> We sincerely thank the reviewer for pointing out the inconsistency between the figure and the textual description regarding the use of transcripts at inference time. The confusion was caused by an incorrect sentence in Sec. 3.1 (line 196 in the original submission) that mistakenly suggested the use of transcripts at test time. We have updated the paper version to remove the wrong statement
>
>  “2) At inference time, given only the observed features $X_{obs}$​ and the video transcript Y,”
> and replace it with
>
> “2) At inference time, given only the observed features $X_{obs}$ and the learned class tokens E.”
>
> TbLTA does **not** require ground-truth transcripts at inference time, as Fig. 2 is intended to show. In the revised version, we believe the training flow is now presented more clearly, since we explicitly separate the training and inference pipelines to reduce visual complexity and avoid misunderstandings (with training and inference flows depicted using distinct arrow styles/colors).
>
> Transcripts are used **only during training** as a weak supervision signal to (1) generate frame-level pseudo-labels via the temporal alignment module, (2) enforce sequence consistency through the CTC loss, and (3) provide semantic guidance via cross-modal attention. At test time, the model observes only the visual features of the partial video $X_{obs}$​ and the class tokens E learned during training; the anticipation decoder operates on their concatenation $[E∥Xobs]$ and its learned queries to produce future action sequences and durations, without any access to ground-truth transcripts.
> To further improve clarity on this important point, we explicitly reiterate in the abstract, introduction, Sec. 3.1, and in the captions of Fig. 1 and Fig. 2 that transcripts are used only during training and are not required at inference. **We sincerely apologize for the confusion our incorrect wording may have caused.**
>
>
> **Closing Note:** We thank the reviewer for the time invested in reviewing our work and for the concrete suggestions regarding figure clarity, motivation for transcripts, and consistency between text and diagrams. These comments led us to correct the inference misunderstanding and substantially improve the presentation. We sincerely hope these points will allow you to revisit your score. We remain fully open to any further questions, suggestions, or discussion.

---

### Official Review · Reviewer_24ux · 2025-11-01

**Soundness:** 2
**Presentation:** 2
**Contribution:** 2
**Rating:** 4
**Confidence:** 4

**Summary:**

This paper presents TbLTA, a novel weakly supervised framework for long-term action anticipation (LTA) that relies solely on video transcripts instead of dense frame-level annotations. The method uses a temporal alignment module (ATBA) to generate frame-level pseudo-labels from transcripts, which are then used to train an encoder-decoder model with a segmentation head and an anticipation decoder. The model also incorporates cross-modal attention between textual and visual features. Experimental results demonstrate that TbLTA achieves competitive performance to fully supervised baselines on Breakfast, and partially comparable results on 50Salads and EGTEA.

**Strengths:**

1. **Clear motivation:** This is the first work to address long-term action anticipation in a weakly supervised setting, which can avoid labor-intensive frame-level annotations.
2. **Tailored objective design:** The integration of multiple loss components (alignment, CTC, CRF, and duration loss) is tailored for weakly supervised scenario. The ablation studies confirm the benefit of each component, particularly the CRF loss, which contributes notable improvements (5.3%p in 50Salads, 4.1%p in Breakfast).
3. **Reasonable empirical validation**: Despite relying only on transcript-level ground truths, it achieves comparable performance with supervised fine-tuning methods (overall in Breakfast, partly in 50Salads and EGTEA).

**Weaknesses:**

1. **Strong reliance on ATBA pseudo-labels**: The most critical limitation of this paper is that the overall framework heavily depends on the quality of pseudo-labels generated by the ATBA alignment module. Since this component originates from prior work on weakly supervised action segmentation, much of the supervision signal in TbLTA is inherited from ATBA. This dependency somewhat limits the originality and isolates less of the contribution to the proposed anticipation framework itself.
2. **Limited exploration of supervision quality**: The paper could better analyze the effect of pseudo-label noise or compare against an oracle using ground-truth labels for training. Such experiments would help contextualize the achievable upper bound and clarify how much performance is lost due to weak supervision. Without this, it is difficult to assess whether the method’s gains stem from the framework design or from the pseudo-label quality.

**Questions:**

1. Dependence on ATBA: The method relies heavily on ATBA to generate pseudo-labels. Could the authors clarify how sensitive TbLTA’s performance is to errors in ATBA? For example, what happens if the alignment quality degrades? is the model robust to noisy pseudo-labels?
2. Alternative alignment strategies: Have the authors explored alternative pseudo label generation methods borrowed from weakly-supervised action segmentation methods other than (Xu & Zheng, 2024)?
3. Oracle comparison: Could the authors provide an oracle experiment where the same model is trained with ground-truth frame labels? This would help quantify the performance gap between weakly and fully supervised setups and reveal how much of the loss stems from label noise versus model design.
4. Generalization to other weak signals: Could the proposed framework be extended to other weak supervision sources (e.g., narrations or subtitles) beyond transcripts? This would make the approach more generally applicable.

---

> ### Author Response · Authors · 2025-11-25
> **Answer to Reviewer 24ux (Part 1)**
>
> We thank the reviewer for his positive comments and insightful suggestions.
>
> > W1., W2 and Q1, Q3. Dependence on ATBA, supervision quality. This is an excellent point.
>
> The CTC loss is a critical component that adds robustness to the noisy labels since it does not depend on the quality of ATBA pseudolabels. In addition, the multimodal cross attention is also designed to add robustness by improving the quality of the features and making a kind of semantic alignment. Our ablation study already validated the role of these components independently.
> To directly address the concern about reliance on ATBA, we performed additional experiments aimed at isolating the effect of the pseudo-label supervision:
>
> - **Training with pseudo-labels only.** We trained a variant of TbLTA where we remove both the CTC loss and the multimodal cross-attention, so that the model is supervised exclusively by the ATBA pseudo-labels. This configuration leads to substantially worse anticipation performance, confirming that (i) the pseudo-labels are indeed noisy (as expected from a weakly supervised method) and (ii) a significant part of the performance gain comes from the proposed anticipation architecture and its additional supervision, rather than from the pseudo-labels alone.
>
> - **Oracle supervision with ground-truth labels.**  Following the reviewer’s suggestion, we also trained a “GT w/o TA module” variant in which the segmentation loss uses ground-truth labels instead of ATBA pseudo-labels and the “GT instead of pseudolabels“ variant, where we keep the alignment module as in TbLTA, but replace the pseudo-labels by ground-truth labels only for supervising the anticipation branch..The resulting performance improves only moderately over the pseudo-label-based version, indicating that TbLTA is able to approach the fully supervised upper bound despite relying on weak supervision. This supports the claim that the framework design, rather than pseudo-label quality alone, is responsible for most of the gains.
>
> - **Sensitivity to pseudo-label degradation.** To further quantify the dependence on ATBA, we conducted a sensitivity analysis where we synthetically degraded the quality of the pseudo-labels by disabling some of the ATBA loss terms and keeping only the main video-level loss active. As expected, performance decreases as the pseudo-labels become noisier, but the degradation is gradual rather than catastrophic. This suggests that the combination of CTC loss and multimodal cross-attention indeed provides robustness to pseudo-label noise.
>
>
> > Q2. Alternative alignment strategies:
>
> We thank the reviewer for this valuable question regarding alternative weakly supervised alignment strategies. In this work, we selected ATBA (Xu & Zheng, 2024) primarily because it offers an excellent trade-off between performance and computational efficiency among weakly supervised action segmentation methods **that are transcript-based and have publicly available code**. As reported in Fig. 1 of the ATBA paper, ATBA is substantially faster than competing approaches, which is a critical requirement in our multi-task setting, where alignment and anticipation must be trained jointly and repeatedly.
>
> For instance, TASL [1] (another strongly weakly supervised method with released code) is more than eight times slower than ATBA in our measurements. Incorporating such a computationally demanding backbone into TbLTA would significantly increase training time and hardware requirements, making extensive experimentation impractical.
>
> We also considered more lightweight alternatives such as POC [2], which are computationally attractive but rely on a different weak supervision setting that **does not use ordered transcripts**. Adopting such methods would require substantial changes to our framework (e.g., redesigning the alignment losses and cross-modal supervision), because TbLTA is explicitly built around transcript-based supervision. For these reasons, we focused our analysis on ATBA in the present work. Nevertheless, TbLTA is conceptually compatible with alternative alignment backbones, and exploring non-transcript-based weak signals is an interesting direction for future work.
>
> [1] Lu, Z., & Elhamifar, E. (2021). Weakly-supervised action segmentation and alignment via transcript-aware union-of-subspaces learning. In Proceedings of the IEEE/CVF International Conference on Computer Vision (ICCV).
>
> [2] Lu, Z., & Elhamifar, E. (2022). Set-supervised action learning in procedural task videos via pairwise order consistency. In Proceedings of the IEEE/CVF Conference on Computer Vision and Pattern Recognition (pp. 19903-19913).

---

> ### Author Response · Authors · 2025-11-25
> **Answer to Reviewer 24ux (Part 2)**
>
> > Q4. Generalization to other weak signals:
>
> We thank the reviewer for this inspiring comment. Narrations and subtitles may be considered a noisy version of the transcripts used in this work since they typically do not provide a complete and precise sequence of actions captured by the videos, but a less fine-grained version, and may need to be matched with action labels used for evaluation. Moreover, subtitles are typically temporally aligned with the video, whereas narrations are often asynchronous, which introduces additional challenges. While transcripts could in principle be derived from such signals using VLMs or LLMs—a direction we find very exciting—this lies beyond the scope of the present work, whose goal is to take a first step from full supervision to weak supervision using exact transcripts. Leveraging narrations or subtitles would represent an additional move towards even weaker and more realistic supervision, which we leave for future research.
>
>
> **Closing Note:** We thank the reviewer for the careful reading and constructive comments on supervision quality, oracle comparisons, and the dependence on ATBA. We have expanded our ablations and clarified the role of pseudo-labels and transcript-aware supervision accordingly. We sincerely hope these points will allow you to revisit your score. We remain fully open to any further questions, suggestions, or discussion.

---

> > ### Comment · Reviewer_24ux · 2025-11-27
> >
> > I sincerely thank the authors for their effort in conducting additional experiments and providing detailed comments. While some of my concerns have been resolved, I find this paper lacking sufficient technical novelty, as other reviewers also raised. Therefore, I will retain my initial score.

---

### Author Response · Authors · 2025-11-25
**Official Comment by Authors (Part 1)**

We would like to thank all reviewers (R1: 24ux, R2: kytL, R3: 6poY, R4: i3xn) for the time and effort dedicated to reviewing our work. We are grateful that several reviewers (R1, R3, R4) highlighted important strengths of our method, including its **clear motivation** to **eliminate the need for expensive frame-level boundary annotations** in dense long-term action anticipation, its **tailored multi-objective design**, and its empirical performance, which is **competitive with fully supervised methods** on standard benchmarks. We also appreciate that reviewers (R1, R2, R3) emphasized the **value of our ablation studies in demonstrating the effectiveness of the proposed losses and modules.**

A key source of confusion for R2 and R4 was an incorrect sentence in Sec. 3.1 (line 196), which incorrectly suggested that transcripts are used at inference time. We sincerely apologize for this mistake. To clarify: **TbLTA uses transcripts only during training** as a weak supervision signal to (1) generate frame-level pseudo-labels via the temporal alignment module, (2) enforce sequence consistency through the CTC loss, and (3) provide semantic guidance via multimodal cross-attention. At test time, the model observes only the visual features of the partial video $X_{\text{obs}}$ and the learned class tokens $E$; the anticipation decoder operates on $[E \Vert X_{\text{obs}}]$ and its queries to predict future actions and durations, **without any access to ground-truth transcripts during inference.** We have corrected the text on line 196 and improved the figures (notably Fig. 2) to make this training–inference distinction more explicit throughout the revised version.


Given this misunderstanding, some reviewers have questioned the novelty of our method, especially the lack of conceptual advances. Others have questioned the dependence on pseudolabels. Although we address these points individually, we view them as important for the broader discussion.

> Lack of conceptual advances.

We want to clarify here that our method is novel in scope, as it is, to the best of our knowledge, the first approach that alleviates the burden of dense frame-level annotation for dense LTA by using video transcripts during training. In doing so, it establishes the first strong baseline for **weakly supervised dense long-term anticipation**. Our contribution is threefold.

**(i) New problem setting and baseline.** To the best of our knowledge, TbLTA is the first framework to tackle dense long-term action anticipation trained **exclusively** from transcripts (ordered action lists without any temporal boundaries or future labels), thereby eliminating the need for frame-level annotations and establishing the first transcript-only supervision baseline on Breakfast, 50Salads, and EGTEA.

**(ii) Novel integration of weak alignment and anticipation.** Methodologically, we go beyond simply “adding transcripts”: we employ a weakly supervised temporal alignment stage to generate soft pseudo-labels for the video and reuse them in a unified multi-task objective that jointly optimizes alignment, TAS, and LTA.

**(iii) New transcript-aware architectural components.** Beyond reusing other modules, we introduce a gated, locally masked multimodal cross-attention mechanism that injects transcript semantics into video features only around the aligned temporal neighborhood, and we apply a CTC loss on encoder frame logits over the full sequence to enforce global transcript consistency while leaving future uncertainty to the anticipation decoder. These components have not been used before in the context of dense LTA, nor combined in a single framework with this objective. Our key conceptual contribution is to show that transcripts can be exploited not only to generate pseudo-labels, but also to make the system more robust to noisy labels by providing additional structured supervisory signals.

---

> ### Author Response · Authors · 2025-11-25
> **Official Comment by Authors (Part 2)**
>
> > Dependence on the temporal alignment module.
>
> We thank the reviewers for this comment. We have added a common ablation table (Table 1 in this rebuttal) on Breakfast that directly targets these concerns and includes the following variants:
>
> **- TbLTA (ours)**
>
> **- GT w/o TA module (No temporal alignment module):** we completely remove the temporal alignment module and train the segmentation and anticipation branches directly with ground-truth frame labels, thus isolating the benefit of having an explicit alignment stage. This is the oracle comparison requested by R1/R3.
>
> **- GT instead of pseudolabels:** we keep the alignment module as in TbLTA, but replace the pseudo-labels by ground-truth labels only for supervising the anticipation branch. This preserves the architecture but removes pseudo-label noise in the anticipation loss.
>
> **- w/o cross-att., CTC:** we remove our two main transcript-aware modules (multimodal cross-attention and CTC); transcripts are then only used indirectly via pseudo-labels. This variant shows how much our transcript-aware regularization contributes on top of pseudo-label supervision. In this case, we show results for TbLTA w/o cross-att., CTC, for GT w/o TA module, cross-att., CTC, and for GT instead of pseudolabels w/o cross-att., CTC.
>
> **- degenerate-TA module (only $L_{\text{vid}}$​):** following the ATBA paper, we synthetically degrade the alignment by keeping only the main video-level loss $L_{\text{vid}}$. In their segmentation ablations, Xu & Zheng show that using only L_{\text{vid}} leads to weaker performance than the full three-loss configuration, indicating that the resulting segmentations (and thus pseudo-labels) are noticeably less accurate.
>
> **Table 1. Ablation study for BF datasets (obs =  $\alpha$, pred= $\beta$).**
> ## MEAN
> | Model  for $(\alpha, \beta)$   | (0.2, 0.1) | (0.2, 0.2) | (0.2, 0.3) | (0.2, 0.5) |(0.2, 0.1) | (0.2, 0.2) | (0.2, 0.3) | (0.2, 0.5) | Avg. |
> |-------------------------------|----------------|--------|--------|--------|----------------|--------|--------|--------|------|
> | TbLTA                         | 28.9           | 25.6   | 24.6   | 21.8   | 38.4           | 35.1   | 31.9   | 28.7   | 29.4 |
> | w/o cross-att., CTC           | 25.7           | 23.0   | 20.5   | 18.5   | 37.0           | 33.0   | 29.3   | 25.8   | 26.6 |
> | GT w/o TA module              | 29.0           | 26.3   | 22.4   | 20.1   | 39.3           | 36.4   | 31.4   | 26.3   | 28.9 |
> | GT w/o TAmod, cross-att., CTC | 27.6           | 24.9   | 22.8   | 19.0   | 38.5           | 34.4   | 31.3   | 25.7   | 28.0 |
> | GT instead of pseudolabels    | 28.6           | 27.6   | 22.9   | 20.8   | 41.4           | 36.4   | 32.4   | 28.8   | 29.9 |
> | GT and w/o cross-att., CTC    | 27.7           | 24.2   | 22.7   | 20.1   | 37.9           | 35.8   | 31.4   | 28.1   | 28.5 |
>
> | -------------------------------------------------------------------------------------------------------------------------------------------------------------------------|
> ## TOP1
> | Model FOR $(\alpha, \beta)$   | (0.2, 0.1) | (0.2, 0.2) | (0.2, 0.3) | (0.2, 0.5) |(0.2, 0.1) | (0.2, 0.2) | (0.2, 0.3) | (0.2, 0.5) | Avg. |
> |-------------------------------|----------------|--------|--------|--------|----------------|--------|--------|--------|------|
> | TbLTA                         | 37.2           | 33.0   | 31.7   | 30.5   | 45.7           | 41.9   | 39.1   | 38.3   | 37.2 |
> | w/o cross-att., CTC           | 31.9           | 28.5   | 27.5   | 25.3   | 43.4           | 39.1   | 35.9   | 33.8   | 33.2 |
> | GT w/o TA module              | 45.6           | 41.5   | 40.5   | 39.3   | 55.5           | 49.6   | 46.3   | 46.2   | 45.6 |
> | GT w/o TAmod, cross-att., CTC | 41.4           | 38.0   | 37.5   | 36.9   | 50.6           | 45.9   | 43.0   | 42.7   | 42.0 |
> | GT instead of pseudolabels    | 41.9           | 37.5   | 35.3   | 32.6   | 49.3           | 45.2   | 42.6   | 41.1   | 40.7 |
> | GT and w/o cross-att., CTC    | 39.7           | 35.7   | 33.4   | 30.8   | 47.7           | 43.5   | 40.6   | 39.8   | 38.9 |
>
> | -------------------------------------------------------------------------------------------------------------------------------------------------------------------------|
>
> First, Table 1 shows that moving from pseudo-labels to GT alignment yields only a modest improvement: the **GT instead of pseudo-labels** oracle improves over TbLTA by about **+0.8 points** in mean and **+3.5 points in top-1** on average. This is consistent with the fact that our performances are competitive with those of fully supervised methods.

---

> ### Author Response · Authors · 2025-11-25
> **Official Comment by Authors (Part 3)**
>
> Second, the ablation confirms that our two key modules are necessary and beneficial even under oracle supervision. When we remove cross-modal attention and the CTC loss, performance consistently degrades. Comparing **GT instead of pseudolabels** **with GT w/o cross-att., CTC**, we observe a drop of roughly **1.4 mean** and **1.8 top-1 points** despite having perfect frame-level labels. In the realistic weakly supervised setting, the effect is larger: **w/o cross-att., CTC loses** about **2.5 mean** and **4 top-1 points** compared to TbLTA. These results directly address the concern that our gains would come only from ATBA: even when ATBA (or GT) provides good labels, the multimodal cross-attention and CTC objectives contribute to additional improvements.
>
> Finally, the **GT w/o TA module** acts as a fully supervised baseline where the segmentation branch and the anticipation branch are trained directly with ground-truth frame labels, i.e., without any weakly supervised alignment. The fact that TbLTA remains competitive with this fully supervised oracle further contextualizes the performance gap between weak and full supervision and addresses R1’s request for an upper bound analysis.
>
> **Overall, a more accurate temporal alignment primarily sharpens short-horizon predictions, whereas our transcript-aware supervision (cross-attention and CTC) is what enables TbLTA to remain competitive over long anticipation horizons, even under weaker alignment regimes.**

---

> > ### Author Response · Authors · 2025-11-26
> > **Official Comment by Authors (Part 4)**
> >
> > To further analyze the dependence on ATBA, Table 2 in part 4 focuses on the **quality of the pseudo-labels** produced by the alignment module and how TbLTA behaves when this quality is deliberately degraded:
> >
> > **Table 2. Ablation study in terms of pseudo-labels for BF datasets (obs = $\alpha$, pred= $\beta$).**
> > ## MEAN
> > | Model  for $(\alpha, \beta)$   | (0.2, 0.1) | (0.2, 0.2) | (0.2, 0.3) | (0.2, 0.5) |(0.2, 0.1) | (0.2, 0.2) | (0.2, 0.3) | (0.2, 0.5) | Avg. |
> > |-------------------------------|----------------|--------|--------|--------|----------------|--------|--------|--------|------|
> > | TbLTA                         | 28.9           | 25.6   | 24.6   | 21.8   | 38.4           | 35.1   | 31.9   | 28.7   | 29.4 |35.1                       | 31.9                       | 28.7                       | 29.4 |
> > | degenerate-TA | 26.5                       | 24.3                       | 21.8                       | 20.1                       | 37.4                       | 34.4                       | 31.4                       | 27.7                       | 27.9 |
> > | single-stage                 | 11.3                       | 9.3                        | 8.6                        | 9.1                        | 14.6                       | 11.6                       | 11.4                       | 12.4                       | 11.0 |
> >
> >
> > | ---------------------------------------------------------------------------------------------------------------------------------------------------|
> >
> > ## TOP1
> > | Model  for $(\alpha, \beta)$   | (0.2, 0.1) | (0.2, 0.2) | (0.2, 0.3) | (0.2, 0.5) |(0.2, 0.1) | (0.2, 0.2) | (0.2, 0.3) | (0.2, 0.5) | Avg. |
> > |-------------------------------|----------------|--------|--------|--------|----------------|--------|--------|--------|------|
> > | TbLTA                        | 37.2                       | 33.0                       | 31.7                       | 30.5                       | 45.7                       | 41.9                       | 39.1                       | 38.3                       | 37.2 |
> > | degenerate-TA (only $L_{\text{vid}}$) | 39.9                       | 32.4                       | 31.3                       | 30.6                       | 49.3                       | 43.1                       | 38.5                       | 38.1                       | 37.9 |
> > | single-stage                 | 15.2                       | 13.2                       | 14.3                       | 18.6                       | 22.2                       | 19.1                       | 20.7                       | 23.8                       | 18.4 |
> >
> > | ------------------------------------------------------------------------------------------------------------------------------------------------------------------|
> >
> > **The degenerate-ATBA (only $L_{\text{vid}}$)** experiment explicitly degrades the temporal alignment module by keeping only one of ATBA’s three losses. Despite this degradation, our framework remains competitive: in Table 2, degenerate-ATBA performs clearly better than the simple single-stage baseline, but it is still about 1.5 mean points below TbLTA on average. This shows that TbLTA does not rely on near-perfect pseudo-labels: even when the alignment quality is deliberately weakened according to the regime identified by ATBA itself, our transcript-aware modules (cross-attention and CTC) provide additional robustness and prevent the LTA performance from collapsing.
> >
> > **Take-away w.r.t. reviewers’ concerns.**
> >
> > Together, Tables 1 and 2 provide: (a) an oracle comparison with ground-truth labels, quantifying the gap between weak and full supervision (R1/R3), (b) evidence that our cross-attention and CTC losses contribute, addressing doubts about conceptual contribution (R2/R3), and (c) an explicit stress test in a degraded-alignment regime.
> >
> > For clarity and to highlight their importance, we have chosen to present the new ablation tables directly within this rebuttal. We will incorporate these tables into the supplementary material or the revised manuscript, as appropriate.
> >
> > **We hope that these clarifications and additional experiments address the reviewers’ main concerns, and we respectfully ask the reviewers to reconsider our contribution in light of this revised perspective.**

---

### Note · Authors · 2026-03-03

I have read and agree with the venue's withdrawal policy on behalf of myself and my co-authors.

---

### Meta-Review · Area_Chair_wSyk · 2026-01-10

**Summary:**

The reviewers raised several concerns, including marginal novelty, complex design, unclear justification, confusing writing, and limited analyses/experiments. The authors’ rebuttal addresses some of them, but fails to fully resolve the issues of marginal novelty and complex design. While the authors demonstrated that the combination of eight loss terms with multi-stage training improves performance, AC also finds that most of the proposed terms and techniques are not new, and multi-stage training with heuristics complicates reproducibility. Siding with the reviewers’ overall ratings, AC recommends rejection.

**Reviewer Concerns:**

The rebuttal clarified the motivation for using transcripts and also corrected the confusion in the manuscript by noting that transcripts are used only in training. The authors also addressed the issue of limited analyses and experiments by providing ablation studies and sensitivity analyses. However, the concerns about marginal novelty and complex design, pointed out by most reviewers, still remain.

**Reviewer Scores:**

Reviewer 24ux retained the original rating of 4 after discussion.
Reviewer kytL would have changed the score from 2 to 3.
Reviewer 6poY would have changed the score from 4 to 6.
Reviewer i3xn would have changed the score from 2 to 4.

---

### Decision · Program_Chairs · 2026-01-26

Reject